# The Confusion is Real: GRAPHIC – A Network Science Approach to Confusion Matrices in Deep Learning

**Johanna S. Fröhlich** *johanna.froehlich@fau.de*
*Friedrich-Alexander-Universität Erlangen-Nürnberg*
*Erlangen, Germany*

**Bastian Heinlein** *bastian.heinlein@fau.de*
*Friedrich-Alexander-Universität Erlangen-Nürnberg*
*Erlangen, Germany*
*Technical University of Darmstadt*
*Darmstadt, Germany*

**Jan U. Claar** *jan.u.claar@fau.de*
*Friedrich-Alexander-Universität Erlangen-Nürnberg*
*Erlangen, Germany*

**Hans Rosenberger** *hans.rosenberger@fau.de*
*Friedrich-Alexander-Universität Erlangen-Nürnberg*
*Erlangen, Germany*

**Vasileios Belagiannis** *vasileios.belagiannis@fau.de*
*Friedrich-Alexander-Universität Erlangen-Nürnberg*
*Erlangen, Germany*

**Ralf R. Müller** *ralf.r.mueller@fau.de*
*Friedrich-Alexander-Universität Erlangen-Nürnberg*
*Erlangen, Germany*

**Reviewed on OpenReview:** *https://openreview.net/forum?id=UP9bx1WJwR*

## Abstract

Explainable artificial intelligence has emerged as a promising field of research to address reliability concerns in artificial intelligence. Despite significant progress in explainable artificial intelligence, few methods provide a systematic way to visualize and understand how classes are confused and how their relationships evolve as training progresses. In this work, we present GRAPHIC, an architecture-agnostic approach that analyzes neural networks on a class level. It leverages confusion matrices derived from intermediate layers using linear classifiers. We interpret these as adjacency matrices of directed graphs, allowing tools from network science to visualize and quantify learning dynamics across training epochs and intermediate layers. GRAPHIC provides insights into linear class separability, dataset issues, and architectural behavior, revealing, for example, similarities between flatfish and man and labeling ambiguities validated in a human study. In summary, by uncovering real confusions, GRAPHIC offers new perspectives on how neural networks learn. The code is available at https://github.com/Johanna-S-Froehlich/GRAPHIC.

# 1 Introduction

Neural networks (NNs) pave the way for automated decision-making in areas that have previously eluded automation due to their high complexity, e.g., self-driving cars (Di Feng et al., 2021) and medical diagnostics (Zhou et al., 2021). Because NNs are usually perceived as *black boxes*, they are often not considered trustworthy (Guo, 2020; Zhang et al., 2021; von Eschenbach, 2021); therefore, the area of explainable artificial intelligence (XAI) has seen a steep rise of interest in recent years (Mersha et al., 2024). By understanding the learning dynamics of NNs, XAI does not only increase trust in NNs, but can also be utilized to improve model performance (Chefer et al., 2022; Yan et al., 2015) or identify dataset issues.

Explainability methods can be applied at different stages of model development: during model design, training, or inference. While inherently explainable models, e.g., Letham et al. (2015) and Lakkaraju et al. (2016), are important especially in high risk applications (Rudin, 2019), these *ante-hoc* methods are not yet feasible for certain tasks (Singh et al., 2024; Atrey et al., 2025; Mumuni & Mumuni, 2025) and cannot be extended to other preexisting models. For this reason, *post-hoc* explainability methods, which aim to analyze models after training without requiring changes to architecture or data, have become the dominant approach.

Unfortunately, many of these post-hoc approaches are limited in scope and offer no insights into the model's overall state. *Local* explainability methods illustrate decisions for individual samples and, for example, use visualizations to gain insights into which features of an image were most relevant for the eventual prediction of an NN (Selvaraju et al., 2020; Wang & Wang, 2022; Bach et al., 2015). Other methods focus on decision regions and their boundaries: Ribeiro et al. (2016) approximated a complex model's decision boundary locally with an interpretable linear model to explain individual predictions, and Karimi et al. (2019) created adversarial examples to identify these boundaries on a more global level. Both approaches rely on explanations derived from individual samples, which may be flawed due to labeling errors or sampling bias and persist in commonly used datasets (Northcutt et al., 2021).

In general, understanding NNs globally, i.e., how and what they learn, is often more relevant than understanding single decisions. Therefore, *global* explainability methods like concept activation vectors (Kim et al., 2018) and, more recently, concept activation regions (Crabbé & van der Schaar, 2022) have been proposed to understand how *concepts*, such as whether an image includes stripes, are distributed in the feature space of NNs and how they relate to each other. Building on this idea, Rigotti et al. (2022) incorporated such predefined concepts directly into the design of NNs. A major limitation of these approaches lies in their reliance on human-defined concepts, which may not align with the abstract representations actually learned by the network, potentially leading to incorrect interpretations or oversimplified conclusions.

Another branch of global explainability focuses on analyzing the structure and complexity of the feature representations within NNs. For instance, Valeriani et al. (2023), Kornblith et al. (2019), and Ansuini et al. (2019) tried to explain the organization of the feature space and quantify the *complexity* of its representations. In most of these methods, the relationship between individual samples is exploited to gain insights into the global behavior and structure of NNs; however, some are only applicable to transformer models.

So far, few explainability methods have been proposed to understand the feature representations and the training process on a class level utilizing *network science*. As we have seen, existing approaches focus on individual samples, making them vulnerable to labeling noise and dataset artifacts, or rely on predefined concepts, which can influence the explanations. In contrast, our explainability approach **G**raph-based **R**epresentation and **A**nalysis of **P**redictions and **H**idden-layer **I**nterpretability via **C**onfusion (GRAPHIC), introduces a global explainability approach based on structural patterns of class confusions. One major obstacle to class-level understanding is that there is no straightforward way to construct confusion matrices (CMs) for hidden layers. CMs indicate how often certain class pairs are mistaken for each other. We suggest employing a simple linear classifier (LC) according to Alain & Bengio (2016) on the feature representations in the hidden layers to generate CMs. Rather than only training the LCs on true labels, we also train them on the labels predicted by the model. This not only sheds light on the feature space and the linear separability of classes, but also gives an accurate understanding of the class learning over the training process without relying on individual samples. These CMs are then analyzed as graphs, enabling us to

trace how class-level structure evolves across layers and training epochs. Unlike methods that rely on predefined concepts or feature attribution, GRAPHIC draws insights directly from data-driven structure, without manual intervention.

Our approach uncovered several important phenomena. In early training, a few dominant classes emerged as confusion hubs, and the order of training data – not just model initialization – shaped early predictions. Over time, semantically meaningful communities (e.g., animals or trees) formed, while confusion between groups decreased. We also identified dataset-specific biases and labeling challenges. For example, the network used seasonal color cues to distinguish tree species, revealing a bias correctable by more diverse data. Finally, we observed that while convolutional neural networks (CNNs) gain linear separability steadily, visual transformers show a decline in separability across early decoders. Our claims are supported by empirical evidence on a CNN and a transformer using two image datasets, as detailed in Section 5.

The main contributions of this work are three-fold:

1. We propose the novel architecture-agnostic analysis approach GRAPHIC, in which we generate CMs for intermediate layers of modern NNs using LCs. The LCs are trained with a custom cross-entropy loss function, integrating both the true and predicted labels. Then, we interpret these CMs as adjacency matrices of graphs and employ two standard methods from network science to analyze the resulting graphs. We demonstrate how these methods can be leveraged to improve our understanding of NN training.

2. We also use GRAPHIC and its visualization ability to discuss and analyze datasets and identify issues that are hidden behind the large number of classes but can be easily spotted using our proposed graph representation. Additionally, we validate these findings with a human study.

3. As a third contribution, we observe the linear separability of intermediate layers. Insights into this property emerge naturally from our methodology. While this idea is not new in general, we observe a decrease in linear separability in early decoders of the analyzed visual transformer.

## 2 Related Work

**Class-Level Explainability Approaches.** While most explainability approaches fall into either local or global categories, few methods address the intermediate level of analysis, where insights into the learning dynamics and the predictions of NNs are gained on a class level from, e.g., CMs. A prominent example is the work by Hinterreiter et al. (2020), which visualizes CMs over time to reveal hierarchical class structures for the final network layer. Other methods use the idea of class hierarchy in order to improve model performance (Yan et al., 2015; Bilal et al., 2018). These methods lack the graph visualization and use of network science for an internal data-driven analysis. The closest related approach is *Confusion Graph* (Jin et al., 2017), which constructs graphs from CMs and applies community detection algorithms from network science to identify groups of classes that are often confused with each other. While their visualization technique is closely related to ours, their analysis only focuses on the final layer and the converged model without discussing the evolution of temporal patterns. A major difference is their use of only the top-$\tau$ predicted classes per sample to construct CMs, potentially missing important yet less prominent connections. A more detailed comparison between the approaches is given in Appendix A.1.

**Dataset Visualization.** Visualizing high-dimensional datasets is a common approach for discovering structure within the data. Methods such as *t-SNE* (Maaten & Hinton, 2008), *UMAP* (McInnes et al., 2018), *LargeVis* (Tang et al., 2016), and *PCA* (Jolliffe, 1986) embed high-dimensional features into a lower-dimensional space suitable for visualization, either of datasets (Pareek & Jacob, 2021) or of their feature representation in NNs (Chan et al., 2018; Alaíz et al., 2020). Most of these methods aim to preserve local relationships, so that points, in this case image vectors or their representation in NNs, that are close in high-dimensional space remain close in the visualization. On the downside, these methods can be sensitive to hyperparameters, and often fail to capture more complex structures (Wattenberg et al., 2016; Böhm et al., 2023). Methods such as *t-SimCNE* (Böhm et al., 2023) go further by integrating contrastive learning

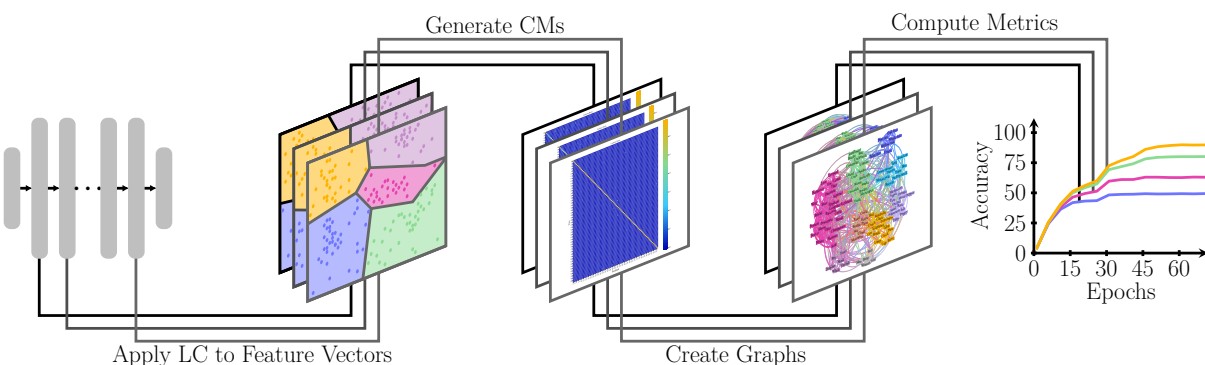

Figure 1: **Proposed analysis workflow.** LCs are trained using feature vectors from hidden layers. The trained LCs are then used to generate CMs on previously unseen feature vectors. Subsequently, these matrices are used to generate graphs that can be analyzed using methods from network science.

with neighbor-embedding techniques, producing visualizations that reveal semantic structure well enough to identify issues and ambiguities within datasets. While we share the goal of revealing these problems, our approach differs fundamentally in how semantically meaningful structures are derived. Instead of visualizing the feature vectors of individual images extracted from an NN, we take a class-based approach: each class is represented as a node in a graph, and edges encode NN confusions. This class-based perspective complements existing approaches that visualize individual images, providing an alternative way to depict datasets and identify their issues.

**Network Science.** Our method also applies network science, which is concerned with the physics of complex systems, i.e., systems with many interacting components (Artime & de Domenico, 2022; Strogatz, 2001). In network science, these systems are modeled as graphs, with components represented as nodes and their relations as edges. Over time, a large set of tools to analyze graphs has been developed for among others community detection (Newman, 2004; Leicht & Newman, 2008), assessing homophily (Newman, 2003b), and understanding structural properties of networks (Newman, 2003a), for various applications (Zhao et al., 2018; Brockmann et al., 2006; Gosak et al., 2018). These tools can also be used beyond complex systems, as long as the underlying data structure can be described as a graph (or *network*).

## 3 Problem Definition

Our objective is to expose and explain how NNs learn by modeling class-level confusion as graphs, applying community detection methods from network science, and interpreting their evolution over the training process. To formalize this, we define a directed graph as a set of nodes connected by weighted, directed edges. Each edge $(i, j)$ from node $i$ to node $j$ carries a weight $A_{i,j} \in \mathbb{R}^1$, with all weights summarized in the square adjacency matrix $\boldsymbol{A}$. In our setting, each node in the graph corresponds to a class in the dataset, and edge weights represent the probability that a sample from class $i$ is predicted as class $j$. The in-degree $\deg^+(i)$ and out-degree $\deg^-(i)$ of a node $i$ correspond to the sum of incoming and outgoing confusions, respectively. To obtain these graphs, we extract the activations $\boldsymbol{h}^{(k)}$ of an intermediate layer $k$ and pass them through an LC, parameterized by weight matrix $\boldsymbol{W}_{\text{lin}}^{(k)}$ and bias vector $\boldsymbol{b}_{\text{lin}}^{(k)}$. The classifier output is given by $\boldsymbol{z}^{(k)} = \text{softmax}\left(\boldsymbol{W}_{\text{lin}}^{(k)}\boldsymbol{h}^{(k)} + \boldsymbol{b}_{\text{lin}}^{(k)}\right)$, which represents the predicted class probabilities at layer $k$. By comparing these predictions with the true labels, we construct a CM for each layer and epoch, interpret them as adjacency matrices of directed graphs and apply community detection algorithms.

---

[1]Scalars, vectors, and matrices are denoted by regular (non-bold), lowercase bold, and uppercase bold letters, i.e., $s$, $\boldsymbol{x}$, and $\boldsymbol{A}$, respectively. Sets are denoted by double-struck letters $\mathbb{A}$ and the cardinality of the set is denoted by $|\mathbb{A}|$.

## 4 Methods

**Proposed Approach.** The analysis workflow is depicted in Figure 1. We begin by training an LC using feature vectors of the hidden layers of the considered NN. Following this, the LC is employed to classify previously unseen feature vectors. The classification results are then used to compute the CMs. Subsequently, these CMs are utilized to generate graphs. Finally, these graphs, and thereby the underlying CMs, are analyzed by applying methods from network science to reveal how and if NNs untangle different classes.

### 4.1 Background: Network Science and Community Detection

**Assortative Mixing in Networks.** The structure of a network reveals the relationships between its nodes. In network science, the tendency of a vertex to connect to a vertex with a shared or opposite characteristic can be analyzed using the so-called assortativity coefficient. Following the formulation from Newman (2003b), we compute the assortativity coefficients of our generated confusion graphs to verify whether predefined concepts like superclasses of a dataset truly align with the structure learned by the model.

As a shared characteristic, concepts relevant to the human understanding are assigned a group label $g \in \{1, \cdots, M\}$, where $M$ is the number of different groups. Two of these concepts are "man-made" and "natural", where *man-made* refers to objects or environments created by humans (e.g., buildings, vehicles), while *natural* refers to those that occur in nature without human intervention (e.g., animals, landscapes).

The group of a class or vertex $c$ is denoted by $G(c)$ and the set of classes belonging to group $g$ as $\mathbb{G}_g$. The assortativity can then be computed utilizing the normalized association matrix $\boldsymbol{E} \in \mathbb{R}^{M \times M}$ that takes the group size $|\mathbb{G}_g|$ into account in accordance to Karimi & Oliveira (2023). With the adjacency matrix $\boldsymbol{C}^{\mathrm{ad}}$ the entries of the association matrix $\boldsymbol{E}$ are computed as (cf. Karimi & Oliveira 2023)

$$E_{u,v} = \frac{1}{|\mathbb{G}_u||\mathbb{G}_v|} \sum_{i \in \mathbb{G}_u} \left( \sum_{j \in \mathbb{G}_v} C_{i,j}^{\mathrm{ad}} \right), \tag{1}$$

with $i, j \in \{1, \cdots, N\}^2$. The matrix $\boldsymbol{E}$ is then normalized elementwise. The assortativity $r$ is computed according to Newman (2003b) as

$$r = \frac{\mathrm{Tr}(\boldsymbol{E}) - \mathbf{1}^{\mathrm{T}} \boldsymbol{E} \boldsymbol{E} \mathbf{1}}{1 - \mathbf{1}^{\mathrm{T}} \boldsymbol{E} \boldsymbol{E} \mathbf{1}}, \tag{2}$$

where $\mathbf{1}$ denotes a vector of all ones and $\mathrm{Tr}(\cdot)$ the trace of a matrix.

**Community Detection Using Modularity.** While grouping classes by some contrived concept can be used to confirm if and when a concept is learned, it is also interesting to examine the community structure that emerges naturally from the network topology. Grouping with a predefined concept relies on external labels, which may not align with intrinsic patterns. One common approach to uncovering these internal communities is through the use of a measure called modularity $Q$. This metric compares the density of edges within communities to the expected density of a random network with the same degree distribution. By maximizing this metric, communities can be found and the quality of the division of a network quantified. The metric is defined by Newman (2004) and Leicht & Newman (2008) as

$$Q = \frac{1}{t} \sum_{i}^{N} \sum_{\forall j: G(i)=G(j)} \left( C_{i,j}^{\mathrm{ad}} - \frac{\deg^+(i) \deg^-(j)}{t} \right), \tag{3}$$

where $t = \sum_{i}^{N} \sum_{j}^{N} C_{i,j}^{\mathrm{ad}}$. We employ the method introduced by Dugué & Perez (2015) to identify community structures. As this method automatically detects the number of communities, it supports intrinsic evaluation by inferring community structure directly from the CMs.

### 4.2 Generating Graphs Using Linear Classifiers

**Training LCs.** We employ LCs to gain insight into both the linear separability of features and the actual "understanding" of an NN under analysis. Contrary to previous works (Alain & Bengio, 2016; Graziani

et al., 2019; Liang et al., 2022), we introduce a novel training approach in which the LC is not only trained on the true labels, but also on the *model predictions*, similar in spirit to knowledge distillation (Hinton et al., 2015). Thus, we define the following *loss function*

$$\mathcal{L}(\boldsymbol{z}^{(k)}, \boldsymbol{y}, \tilde{\boldsymbol{y}}) = \lambda \, \mathcal{L}_{\text{CE}}(\boldsymbol{z}^{(k)}, \boldsymbol{y}) + \left(1 - \lambda\right) \mathcal{L}_{\text{CE}}(\boldsymbol{z}^{(k)}, \tilde{\boldsymbol{y}}), \qquad (4)$$

where $\boldsymbol{z}^{(k)}$ denotes the outputs or predictions of the LC, $\mathcal{L}_{\text{CE}}$ denotes the cross entropy loss, $\boldsymbol{y}$ denotes a one-hot encoded vector containing the true label and $\tilde{\boldsymbol{y}}$ denotes the label predicted by the NN under analysis. The parameter $\lambda \in [0, 1]$ weights the influence of the true labels and the model predictions. Here, we explore the two boundary cases that reveal distinct and complementary insights:

1. $\lambda = 1$ (true labels): The LC is trained only on *ground-truth* labels. Its accuracy directly translates to class separability.

2. $\lambda = 0$ (predicted labels): The LC is trained on the model's current *predictions*. Because the LC receives no information beyond the model output, its accuracy does not exceed that of the model (cf. Appendix A.2); it matches the model's true performance and gives insights into the training process and the class learning. It is therefore preferred for the interpretability investigations.

Intermediate $\lambda$-values yield results that lie between these two cases and are discussed in Appendix A.3. In addition, the impact of training with weight decay is discussed in Appendix A.4.

**Generation of CMs.** LCs are employed to generate CMs. In this paper, we use *normalized* CMs $\boldsymbol{C}$, where the entry $C_{s,t}$ describes the fraction of samples belonging to class $s$ that are classified as class $t$. So the rows of the CMs represent the true labels of the dataset and the columns the labels predicted by the LC. We create CMs for both the training and validation sets. For this, the training set is split into two parts: one is used to train the LCs, while the other is used to compute CMs based on feature vectors that were not seen during LC training. The trained LCs are also applied to the validation set to generate CMs from its feature vectors.

**Generation of Weighted Graphs.** Next, a graph is created for each layer leveraging the previously generated CMs. The graph is described by its adjacency matrix $\boldsymbol{C}^{\text{ad}} \in \mathbb{R}^{N \times N}$, where $N$ is the number of classes. We set $\boldsymbol{C}^{\text{ad}} = \boldsymbol{C} - \boldsymbol{C} \odot \boldsymbol{I}_N$, as we are only interested in the erroneous predictions, where $\odot$ represents the Hadamard product and $\boldsymbol{I}_N$ the identity matrix of size $N \times N$. This leads to a *weighted* and *directed* graph.

## 5 Experiments and Results

We first describe our experimental setup for both the NN training and the training of the LCs in Section 5.1. In Section 5.2 we discuss the evolution of confusion communities and show our graph visualizations. In Section 5.3 dataset issues are uncovered, and we investigate the linear separability of classes in Section 5.4.

### 5.1 Experimental Setup

**Model and Dataset.** Unless specified otherwise, all experiments were conducted on CIFAR-100 (Krizhevsky et al., 2009), a dataset which contains images of 100 classes, each of which belongs to one of 20 superclasses. We used this dataset due to its open availability and low computational requirements, which allow reproducibility even with limited hardware resources. Additionally, we present results on Tiny ImageNet (Le & Yang, 2015), which contains 200 classes to demonstrate the generalization and scalability of our approach to larger datasets and discuss further scalability adjustments in Appendix A.5.

We trained ResNet-50 (He et al., 2016) without pretraining on CIFAR-100 with batch size 1,024 and learning rate 0.003 for 71 epochs with the Adam (Kingma & Ba, 2015) optimizer. For the vision transformer introduced by Saghar Irandoust et al. (2022), which we refer to as EffVit throughout this work for readability, we followed their training setup and used batch size 64, learning rate 0.001 and AdamW (Ilya Loshchilov &

Frank Hutter, 2019) optimizer for both CIFAR-100 and Tiny ImageNet. The model was trained for 1,000 and 341 epochs, respectively. We chose these networks as we are interested in analyzing two different architectures. For CNNs, ResNet-50 serves as a widely recognized baseline and is frequently used in literature, making it a suitable choice (Rangel et al., 2024). As we are also interested in transformer-based models and want to provide easily reproducible results, we trained EffVit as it is accessible to a broad audience. It can be trained in 24h using a single GPU. In all our evaluations we use a single 32 GB NVIDIA V100 GPU.

**LC Implementation.** At every 5 training epochs for ResNet-50 and every 10 for EffVit, one LC was trained on the feature vectors from the model: for ResNet-50, from the output of each block; and for EffVit, from the outputs of five decoder stages. Due to the varying dimensionality of layer outputs in ResNet-50, we used four different learning rates for the LCs: 0.0001, 0.0002, 0.0006, and 0.001. For EffVit, we trained the LCs for both datasets with learning rate 0.01 for each decoder, as the dimensionality of the features is the same. We trained the LCs on 80 % of the images from the training partition, while the remaining 20 % and the validation split were used to compute the CMs for the generation of the graphs. We present results for a single initialization of the LCs and demonstrate their robustness to initialization and hyperparameters in Appendix A.6. The computational overhead introduced by this as well as practical guidelines on choosing which epochs to analyze are discussed in Appendices A.7 and A.8, respectively.

## 5.2 Confusion Communities

In this section we identify confusion communities (CCs) from the introduced graph representation and discuss how they evolve over the training process and through the layers. We are interested in the learning process of NNs, and therefore look at the graphs created with $\lambda = 0$ for the training images, i.e., the LC is trained on the predictions of the NN and evaluated on unseen images from the training set. We identified CCs by maximizing the modularity of the groupings (cf. Section 4.1). The modularities for both ResNet-50 and EffVit are depicted in Appendix A.9, Figures 30 - 34. Figure 2 shows the graph representations for Resnet-50 for epochs 1, 26, and 71 for the final layer, where the CCs are depicted by proximity and color of the nodes. The epochs were chosen to represent three stages of the training; mostly untrained (accuracy 3.14%), mediocre classification performance (accuracy 54.95%) and converged model (accuracy 70.26%). The thickness of the edges indicates their weight, and arrows denote directionality. Graph sparsity and its implications are discussed in Appendix A.10. For a more detailed analysis, full-resolution vector graphics are available in the accompanying GitHub repository. The graphics were created using Gephi (Bastian et al., 2009).

**Early Training Confusion Hubs.** After training for just one epoch, few classes in our representation nodes emerged as hubs in the graph. Hubs are nodes with a large in-degree. As highlighted in Figure 2 in the left plot for ResNet-50 the most predicted classes are computer keyboard, sea and possum. The same is observed for EffVit (cf. Appendix A.11, Figure 38): the hubs here are different to the ones of ResNet-50, as the architecture, number of parameters and initialization are also different. When reversing the order of the dataset and shuffling the dataset with a set seed, we observe that this also changes which classes are predicted most often in the first epoch (cf. Appendix A.11, Figure 40). Since the weight initialization in our case is the same, any variation in predictions here can be attributed solely to the data ordering. This suggests that the order of the training dataset is important for the performance in early stages of the training. It also shows that the initial predictions are not only inherent to the model initialization, but also rely on the initialization of the dataset. These hubs also end up in different CCs for the converged models. This observation also raises the question whether some classes are inherently more difficult to learn than others, which is investigated in Appendix A.12.

*Main Takeaway:* In the first training epoch, a few dominant classes act as confusion hubs, and the dataset ordering – not just model initialization – plays a key role in shaping early predictions.

**Emergence of Semantic Groupings Over Training.** As we continue the training process, in Figure 2 in the right plot, we can see the emergence of clear patterns, where humans, trees, animals, things, etc. are already grouped together. The CCs are still heavily intertwined at this stage. In comparison, for the converged model (bottom plot) we observe a reduced intergroup connectivity, accompanied by an increased

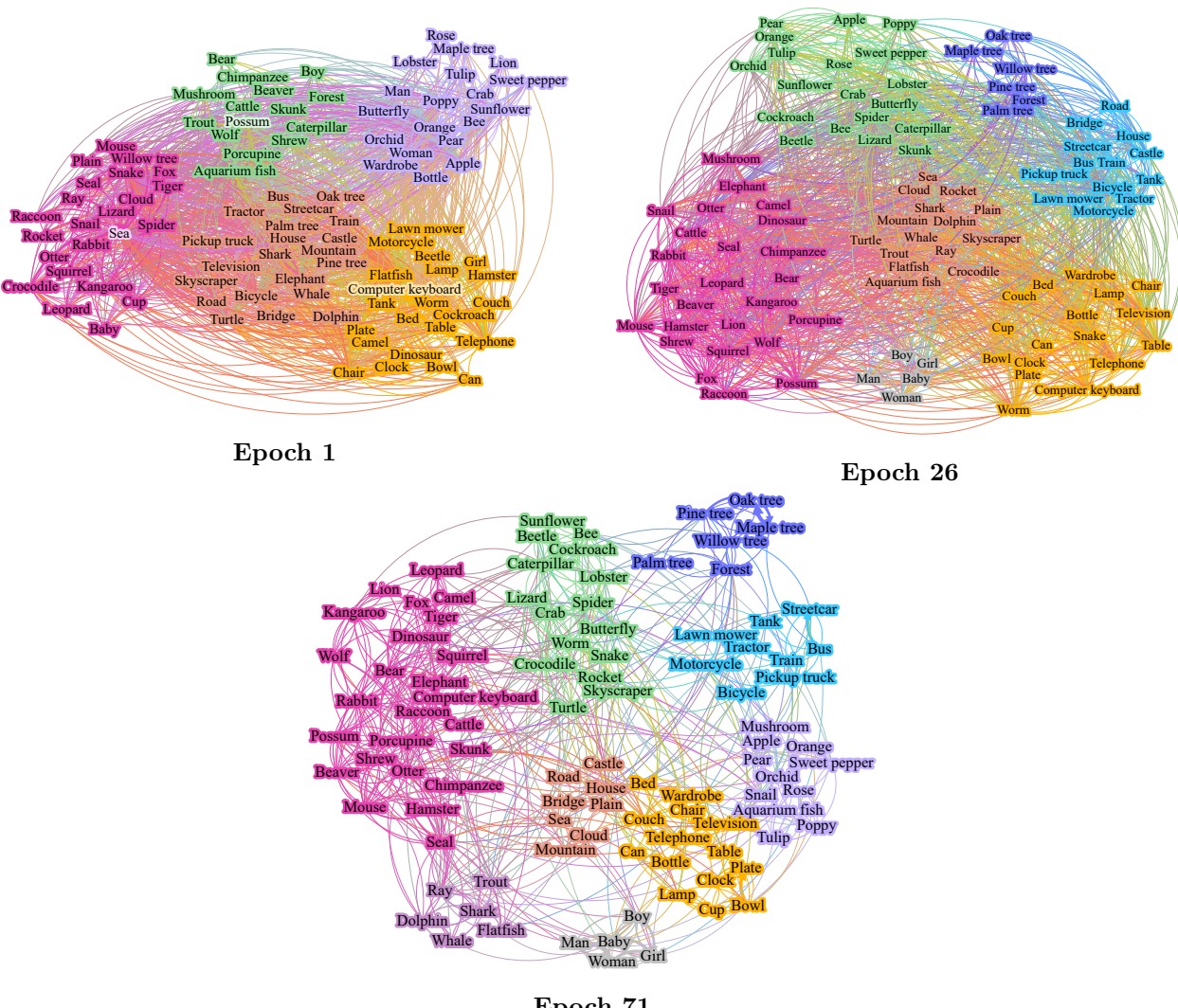

Figure 2: **Confusion evolution of ResNet-50 using the training set.** Visualization of CCs for layer 4 at early (left), intermediate (right), and final (bottom) epochs using our graph representation for the training set.

number of groupings. While this trend arises in all layers of ResNet-50, the difference is in the certainty of the grouping and the intergroup connectivity. In the final training epoch, early layers show more connections, so more confusions, than the final one. The groups are also less refined and the modularity is lower. Evaluating the LC on the validation set leads to similar groups with a higher connectivity between groups; the tree and scenery communities are merged for the final layer in the final epoch. The corresponding graphs are depicted in Appendix A.11, Figure 37. It is also noteworthy that certain classes like oak and maple tree are very often confused with each other. The same is apparent for humans. These strong connections can be interpreted as an indicator for issues in the dataset and are discussed in Section 5.3. A similar behavior is observed for EffVit. The confusion graph evolutions for the training and the validation set of EffVit can be found in Appendix A.11, Figures 38 and 39. After the full training, the graph using the training images in the final decoder is, however, barely interpretable as the accuracy is almost 100 % and there are few connections in the final epoch.

*Main Takeaway:* As training progresses, semantic groupings like animals or trees begin to form, with clearer, more refined communities and reduced intergroup confusion at convergence.

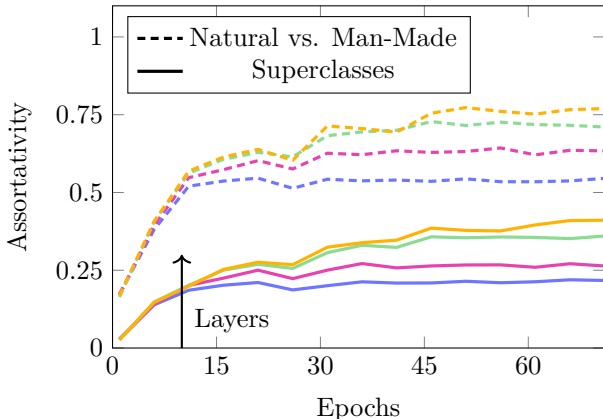

Figure 3: **Layer-wise assortativity over training epochs.** Assortativity computed by superclasses (solid lines) and by natural vs. man-made grouping (dashed lines), for layers 1 through 4 of ResNet-50.

**Comparison to CIFAR-100 Superclasses.** The identified CCs, as depicted in Figure 2, are different from the superclasses, used in CIFAR-100. From a human perspective this is easily explained by the nature of the superclasses. Mammals are, for example, present in several superclasses as small mammals, medium-sized mammals or large carnivores. From a human perspective they all fall into the group mammals. This observation is also confirmed when we analyze the assortativity, as plotted in Figure 3. Following the categorization by Al Musawi et al. (2022), we consider networks with $r > 0.7$ to show high assortativity, those with $0.25 < r < 0.7$ as moderately assortative, and values below $r < -0.25$ as indicative of disassortative mixing. The assortativity for the superclasses through all layers remains relatively low, but still increases with the training and is higher for deeper layers. This means that only a weak assortative pattern can be found in regard to the superclasses. If we split the dataset into two groups, namely natural things and things made by man, we can see that the assortativity after just one training epoch is already fairly high and increases to show a clear assortative structure of the confusion graphs. This can be interpreted as the NN quickly learning to distinguish natural and man-made things, but is also partially due to difference in group sizes. In Appendix A.13 we discuss the influence of the group size in regard to random groupings, where we assign classes to a group at random and compute the assortativity for these random groupings.

*Main Takeaway:* The network quickly learns to distinguish natural from man-made objects, while the pre-defined CIFAR-100 superclasses show only weak alignment with the confusion-based groupings.

### 5.3 Dataset Issues

**Leaf Color Bias in Tree Classification.** GRAPHIC can also be used to analyze potential issues within the dataset, as it highlights which classes cannot be separated by an NN. To get a first idea, we look at the final layer of the converged model (cf. Figure 2 bottom graph). We find that maple trees are often confused with oak trees and vice versa, even though less pronounced. Taking a look at the images it becomes clear that the coloring may influence the network decision. We find that maple trees are often shown in fall with red or yellow leaves, whereas oak trees are mostly colored in green. To test this hypothesis, we changed the color of 10 images, 5 from the class oak tree and 5 from the class maple tree, respectively. An example is depicted in Figure 4, the other images are in Appendix A.14, Figure 46.

The original image is incorrectly classified as oak tree by ResNet-50. With the color change maple tree is correctly identified. The color changes for the oak trees lead to a consistent decrease in the probability of the class oak tree. This suggests a strong relationship between the leaf color and the predicted classes. We show detailed results for the other images in the appendix. This issue could be addressed by adjusting the dataset to include a balanced number of trees from all seasons, so that color becomes less indicative of a specific tree type.

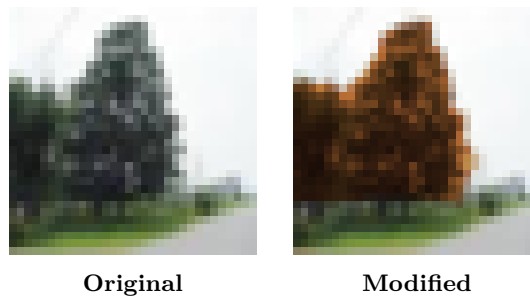

**Original**     **Modified**

Figure 4: **Effect of leaf color on classification.** Example image of a maple tree before (left) and after (right) color manipulation.

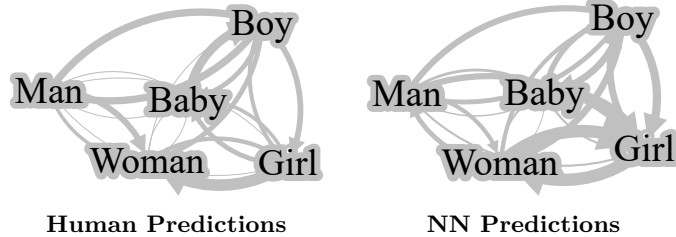

**Human Predictions**     **NN Predictions**

Figure 5: **Human CC created from human and NN predictions.** Visualization of the confusion graph of the human labeling (left) and of the NN predictions (right).

*Main Takeaway:* NNs rely on seasonal leaf color for distinguishing oak and maple trees, indicating a dataset bias that could be mitigated through more diverse, seasonally balanced data.

**Ambiguous Human Class Labels.** A second inconsistency can be found when looking at the classes man, woman, boy, girl and baby. Apparently these classes are hard to distinguish, as the classes, i.e., nodes in the graph, show strong connections and are in one CC. While this makes sense as all images show humans, the strength of the connection leads us to check the images from the validation set. We quickly spot potential issues: a man or woman holding a baby, boys and girls hardly distinguishable due to the poor quality of the images, at what age is a baby a boy or girl and at what age a man or woman?

To further study this issue we enlisted 31 people from ages 21 to 66 to label the human images (man, woman, boy, girl and baby) of the validation set. For more details about the participants refer to Appendix A.15. Exemplary images we consider ambiguous due to high disagreement among participants are depicted in Appendix A.15, Figure 47. The study was split into 5 questionnaires with 100 questions each. Two thirds of the images were different across the five questionnaires. Each participant saw all five questionnaires, which were the same for each participant. One third of duplicate images was added to see whether these images were labeled the same or different as the previous time by each participant. The results for this are in Appendix A.16.

The confusion graph created without duplicate images is depicted in Figure 5. Comparing this to the prediction of the NN the value of the wrong predictions is in general lower, however, the confusion trends are similar. Like the NN humans confuse the classes boy, girl, and baby with each other. This is not surprising, as toddlers could be interpreted as babies, boys or girls. Further, the gender can often only be interpreted using gender stereotypes, like the color of a shirt.

*Main Takeaway:* CCs reveal that classes like boy, girl, and baby are difficult to distinguish – even for humans – due to labeling ambiguity and image quality.

**Contextual Bias in Flatfish Images.** If we take a look at the graphs of earlier layers like layer 2 of ResNet-50, we often see a strong connection between the class flatfish and the class man (cf. Appendix A.11,

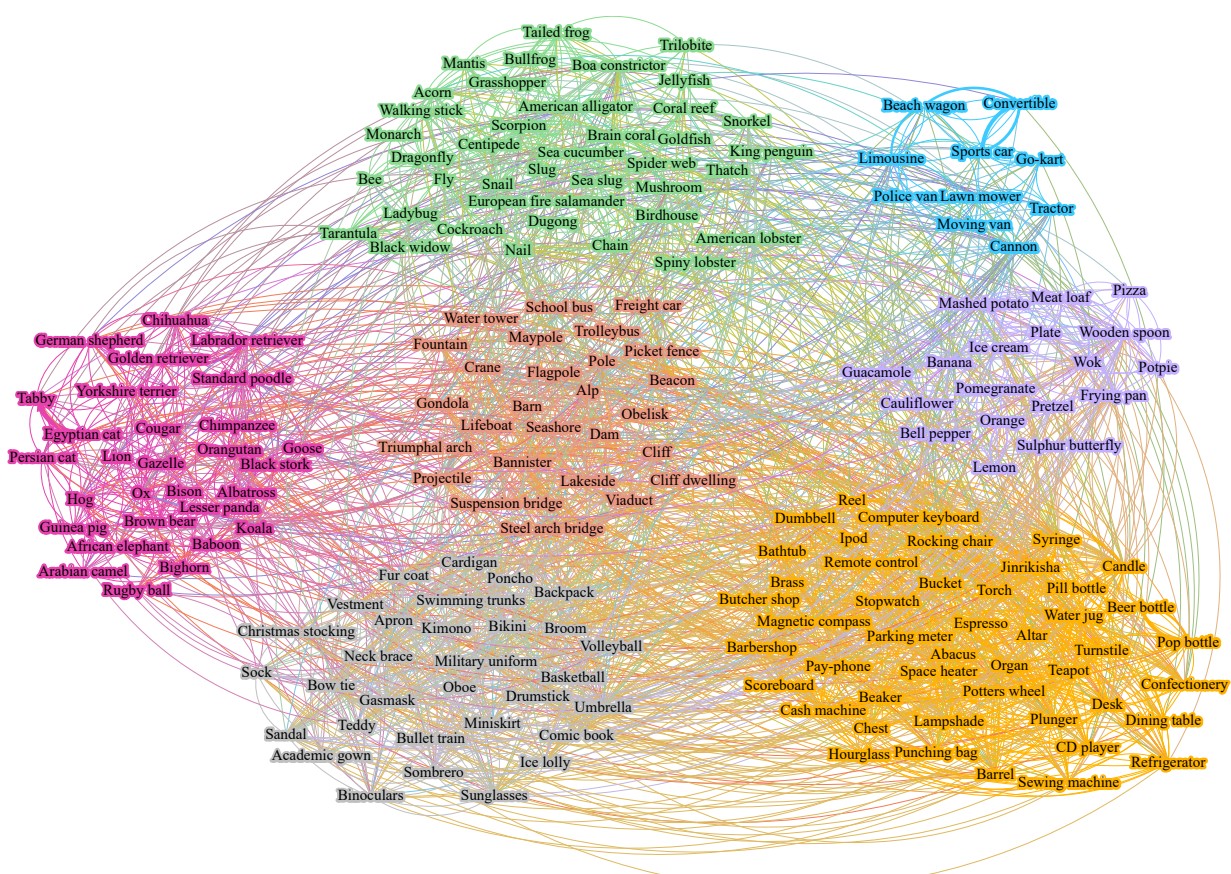

Figure 6: **Confusion graph of EffVit for Tiny ImageNet using the validation set.** Visualization of CCs for decoder 12 at the final epoch using our graph representation for the validation set.

Figure 41). The connection is interesting, as men are not confused with flatfish, but flatfish are confused with men, so the connection is not reciprocal. Flatfish is also sometimes part of the CC of the human classes. When we look at the images of the flatfish class it becomes apparent why flatfish are confused for men. The class contains many images where proud anglers hold up their catch. This issue with the dataset has already been addressed in literature (Wei et al., 2022; Böhm et al., 2023); nevertheless, it highlights how GRAPHIC is able to visualize dataset errors.

*Main Takeaway:* Strong confusion between flatfish and man stems from contextual artifacts in the dataset rather than actual visual similarity.

**Dataset Ambiguities in Tiny ImageNet.** Examining the graph representation for the validation set of Tiny ImageNet (Figure 6), we can observe several clear patterns indicative of dataset issues. For example, convertibles are often predicted as sports cars and tabby and Egyptian cats are confused with each other, but not with Persian cats. If we look at the images of the dataset this not surprising. Many of the sports car images in the dataset are in fact convertibles. This overlap creates ambiguity in the labels. The confusion between tabby and Egyptian cat is also related to the dataset: tabby is not a breed but a description of a fur pattern (Cambridge University Press, 2025), and many Egyptian cats in the dataset show these markings, while Persian cats do not show this pattern.

*Main Takeaway:* Even in larger and more complex datasets, GRAPHIC effectively reveals confusions arising from ambiguous or inconsistent labels rather than model shortcomings.

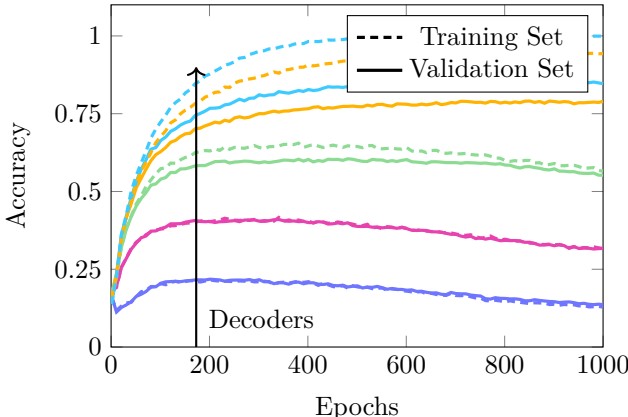

Figure 7: **Linear separability trends in EffVit.** Accuracy of CMs generated by LCs trained on true labels for decoders 1, 3, 6, 9, and 12 of EffVit, shown over the training epochs.

### 5.4 Linear Separability

We analyzed the linear separability of features in EffVit by training LCs on the true labels (i.e., with $\lambda = 1$). The accuracy of these LCs serves as a direct measure of linear separability throughout the network (Alain & Bengio, 2016). According to Alain & Bengio (2016), linear separability is enforced by the final layer and increases over the training epochs and for deeper layers. For ResNet-50, we found the expected trend; it is depicted in Appendix A.2, Figure 11.

As shown in Figure 7, we observe that for EffVit in early training epochs, accuracy increases across all decoders, indicating growing separability. While this trend continues for the later decoders, the early decoders begin to *unlearn* this separability as training progresses. To investigate whether this is tied to model depth or the specific dataset, we trained EffVit variants with 12, 8, and 4 decoders for CIFAR-100 and with 12 decoders for Tiny ImageNet (cf. Appendix A.2, Figures 8, 9 and 10). In all cases, we consistently observe an initial rise, followed by a decline in linear separability in the early decoders. The decoders seem to learn differently from CNNs. This is supported by the observation that visual transformers have to learn locality behavior through training, instead of inherently "knowing" this concept like CNNs (Raghu et al., 2021). This work found that early layers attend both locally and globally, while later layers attend mostly globally, which may be an explanation for the separability behavior.

*Main Takeaway:* EffVit's early decoders initially gain, but then lose linear separability during training. This effect persists across architectures with 4, 8, and 12 decoders, suggesting a possible difference in how visual transformers learn compared to CNNs, where linear separability increases monotonously through training.

## 6 Limitations and Conclusion

GRAPHIC detects dataset-related issues by visualizing how NNs confuse classes. It reveals architectural differences and similarities between NNs and offers insights into the linear separability of features. By providing interpretable, class-based visualizations of the training process, GRAPHIC opens new avenues for debugging and dataset design. Additionally, it draws on concepts from network science to analyze class confusions from a data-driven perspective. While the experiments presented focus on image recognition tasks, GRAPHIC can be extended to any classification task as long as labeled data is available. Exploring extensions in areas like speech classification or text sentiment analysis are promising avenues for understanding NNs in the future. A current limitation is, however, the overhead created by training the LCs. As future work will focus on integrating graph construction directly into the training pipeline, we plan to explore strategies such as reusing LCs pretrained from the previous epoch in order to reduce this overhead and increase the method's accessibility. Additionally, fully leveraging GRAPHIC requires manual interpretation of the results to understand the semantic meaning of the observed confusions. The ability to uncover real confusions has

direct implications for high-stakes applications, such as medical imaging, where understanding systematic errors is crucial to reliable and trustworthy use. GRAPHIC provides an actionable method to guide dataset design, improve model reliability, and enable safer deployment of NNs in critical domains.

**Acknowledgments**

The authors gratefully acknowledge the scientific support and HPC resources provided by the Erlangen National High Performance Computing Center (NHR@FAU) of the Friedrich-Alexander-Universität Erlangen-Nürnberg (FAU). The hardware is funded by the Deutsche Forschungsgemeinschaft (DFG, German Research Foundation). We would also like to thank Amy G. Schol, Peter M. Fröhlich, and Peter R. Fröhlich for their helpful feedback on the manuscript. This work was supported by the DFG under the projects Computation Coding (MU-3735/8-1 and RE 4182/4-1) and GRK 2950 (Project-ID 509922606).

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

# A  Appendix

The appendix is structured as follows: GRAPHIC is compared to existing approaches in Appendix A.1. Appendix A.2 discusses linear separability trends in EffVit and ResNet-50, while Appendix A.3 covers $\lambda$ values between 0 and 1. Training LCs with weight decay is considered in Appendix A.4 and Appendix A.5 provides strategies for scaling the approach to datasets with more classes. The robustness of the LCs is verified in Appendix A.6. At the same time, Appendices A.7 and A.8 provide wall-clock time measurements and practical guidelines on how to choose which layers and epochs to probe. Additional information on modularity is provided in Appendix A.9, while the evolution of sparsity in the graphs is examined in Appendix A.10. Further graph visualizations are depicted in Appendix A.11 and the difficulty of classes is analyzed in Appendix A.12. Appendix A.13 discusses the effect of group size on assortativity. Appendix A.14 analyzes the influence of leaf color on tree classification, Appendix A.15 discusses ambiguous image labeling, and Appendix A.16 presents additional results of the human study.

## A.1  Relation to Existing Explainability and Visualization Methods

GRAPHIC relates to several existing approaches in XAI and model analysis, but differs in its assumptions, scope, and level of supervision. Concept bottleneck models (Koh et al., 2020) are intrinsically interpretable architectures that explicitly decompose prediction into two stages: predicting human-defined concepts and then using these concepts to predict the final class. This requires a dataset, where concept annotations must be available during training. Post-hoc concept bottleneck models have also been proposed (Yuksekgonul et al., 2023), but still rely on predefined concepts. GRAPHIC is a post-hoc method that operates on standard classification datasets and can be applied to any NN without modifying its architecture, affecting its behavior or annotating the existing dataset.

Concept activation vectors (Kim et al., 2018) is a post-hoc interpretability method that defines concepts through user-provided examples: a set of samples that exhibit a given concept and a corresponding set that does not. An LC is then trained on intermediate feature activations to distinguish between these two groups, but not utilized to generate CMs or graphs. Instead, resulting in a concept activation vector that represents the direction of the concept in the feature space. This direction can be used to quantify how sensitive predictions of an NN are to changes along the concept direction, either at the level of individual predictions or in aggregate, for example, by measuring the fraction of samples in a class that are positively influenced by the concept. Again, our approach does not rely on concepts.

Our method could, however, be extended to include this analysis. While we work with true labels and model predictions as labels, we could define concepts as labels to train the LCs as well. With that, one could analyze how these concepts relate to each other, e.g., whether the concepts striped and dotted are often confused with each other. While this would require additional labels, this is interesting for datasets with very few classes, where class confusions can give only few new insights.

t-SNE (Maaten & Hinton, 2008) and UMAP (McInnes et al., 2018) visualize representations by first computing pairwise similarities between samples and then mapping them into a low-dimensional space. When

these similarities are computed in the input space, however, it is not guaranteed that distance measures such as the Euclidean distance in high-dimensional settings are semantically meaningful (Aggarwal et al., 2001). Prior work has identified dataset artifacts, such as the flatfish confusion (Böhm et al., 2023), by looking at where individual images are grouped into classes, but utilized contrastive learning for the representation. In contrast, GRAPHIC operates directly on class-level confusions and therefore does not rely on pairwise image similarities. Furthermore, while t-SNE and UMAP produce one point per image, GRAPHIC analyzes behavior at the class level.

*ConfusionFlow* by Hinterreiter et al. (2020) visualizes how confusions evolve over time directly in the cells of a CM. While this provides a temporal view of misclassifications and gives insights into which classes are how often confused with others, the approach is limited in scalability. The authors note that it is practical for datasets with up to 20 classes, as larger class counts quickly become difficult to visualize due to screen resolution constraints. In contrast, GRAPHIC does not visualize metrics in the CMs directly, but visualizes the confusions as weighted graphs and extends the analysis to intermediate layers of the network. This graph-based formulation allows for the use of community detection and metrics from network science to learn more about class relations and, as shown in the paper, is applicable to datasets with more than 20 classes.

The closest related work to our method is Confusion Graph (Jin et al., 2017). Both methods visualize CMs as confusion graphs and use tools form network science to analyze CCs. However, this formulation differs from GRAPHIC in several fundamental ways. First, restricting the graph construction to the top-$\tau$ predicted classes can lead to confusions being missed, even when they are systematic. An example of this can be seen when looking at their representation of CIFAR-100. By removing weak confusions of the final layer they miss the connection between the classes flatfish and man. While this is also not a dominant confusion in our representation in the converged final layer, it is consistently found in the graph through close inspection. This is also why GRAPHIC is designed for visualizing not only the final layer. As we have discussed in Section 5.3, the class flatfish is part of the human CC for the converged model in layer 2, but not in layer 4, which makes it easier to spot in early layers. Furthermore, the CCs of Confusion Graph are generally smaller than the ones we find, as nodes are less connected overall.

Another difference lies in the directionality of the depicted graphs. While GRAPHIC utilizes directed graphs in an effort to preserve asymmetries in the CMs, Jin et al. (2017) construct undirected graphs, which can lead to non-reciprocal connections being averaged and thus be overlooked.

This effect is visible in the analysis of tree classes. Jin et al. (2017) attribute these confusions broadly to similarities in texture and color. While this explanation is generally valid, it obscures more specific dataset effects. GRAPHIC shows that confusions between trees are especially strong from maple trees to oak trees. As discussed in Section 5.3 this is likely caused by the seasonal bias in the data. When represented as an undirected edge, this asymmetric pattern is averaged, reducing its apparent strength in comparison to other confusions between trees and making the underlying color-driven bias harder to detect.

Finally, as our goal is not solely to visualize confusion patterns in NN, but also to understand the training process, the temporal aspect of GRAPHIC extends the analysis of Confusion Graph in that direction. This allows us to study how class confusions emerge, evolve and in some cases disappear over time, providing additional insights into the learning dynamics of the NN.

## A.2 Accuracy

The linear separability trends (the accuracies of the LCs) for EffVit with 8 and 4 decoders are depicted in Figures 8 and 9, respectively. As discussed, they show a continuous increase in the linear separability over the training process for later decoders, but show an early increase and then a gradual decrease for early decoders. The same trend is observed for EffVit trained on Tiny ImageNet as depicted in Figure 10. This is interesting, as it differs from the linear separability trends in ResNet-50, as plotted in Figure 11. It shows the accuracy of the different layers of ResNet-50 determined by LCs trained on the true labels in comparison to the true accuracy of ResNet-50. Here we see an increase ending in a stagnation, rather than a drop.

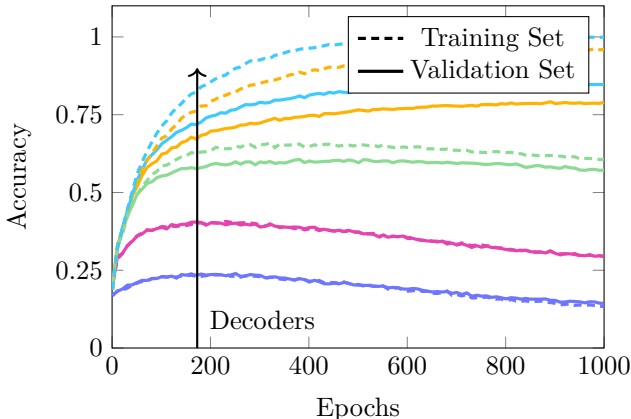

Figure 8: **Linear separability trends in EffVit with 8 decoders.** Accuracy of CMs generated by LCs trained on true labels for decoders 1, 2, 4, 6, and 8 of EffVit, shown over the training epochs.

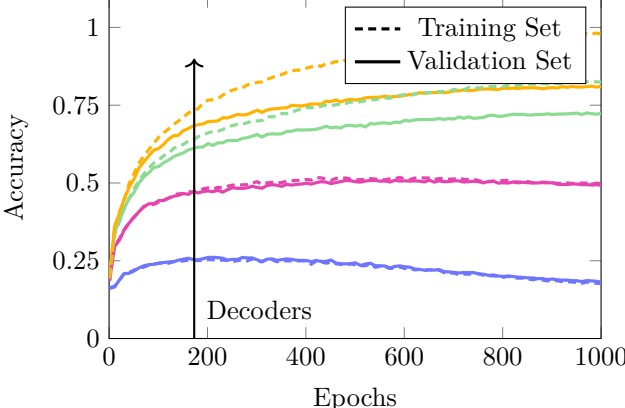

Figure 9: **Linear separability trends in EffVit with 4 decoders.** Accuracy of CMs generated by LCs trained on true labels for decoders 1, 2, 3, and 4 of EffVit, shown over the training epochs.

The accuracy for the LCs trained on the true labels also represents the true potential or the linear separability of the layer outputs at that stage. An LC trained on the true labels is basically a decision maker that is allowed additional training in comparison to the last layer of the NN. This is also why the LC initially outperforms the accuracy of the NN, but converges to a similar accuracy as the model. The final difference in accuracy can be attributed to the split in the training set. Due to this split, the LC is not trained on all images the NN is trained on. In the early training it gives an upper bound of the accuracy. Interestingly, the accuracies for layer 3 and layer 4 are almost the same. This means that instead of using ResNet-50 for the inference phase, one could also train an LC on the features of layer 3 and use the LC to reduce the cost of inference. A similar observation was made by Teerapittayanon et al. (2016).

If the LCs are trained on the predicted labels, they give an accurate representation of the true "understanding" of the model. Figure 12 depicts the layer-wise accuracy created through these LCs. The accuracy of the final layer aligns with the accuracy of the model. The differences, e.g., between epochs 25 and 30, stem from the differing evaluation intervals, as the model's accuracy is recorded at each epoch, while the CMs are evaluated every five epochs.

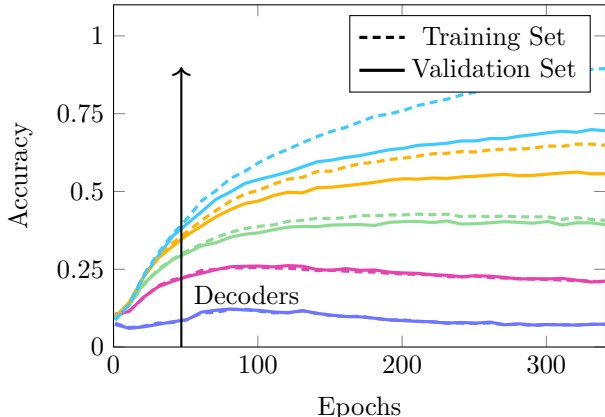

Figure 10: **Linear separability trends in EffVit for Tiny ImageNet.** Accuracy of CMs generated by LCs trained on true labels for decoders 1, 3, 6, 9, and 12 of EffVit, shown over the training epochs.

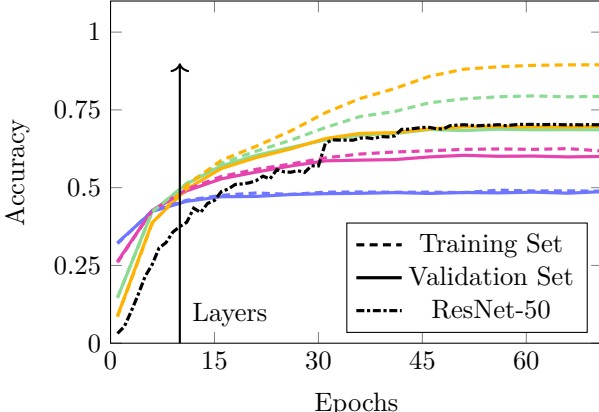

Figure 11: **Linear separability trends in ResNet-50.** Accuracy of CMs generated by LCs trained on true labels for layers 1 to 4, shown over the training epochs. The graph also includes the true accuracy of ResNet-50 as a baseline.

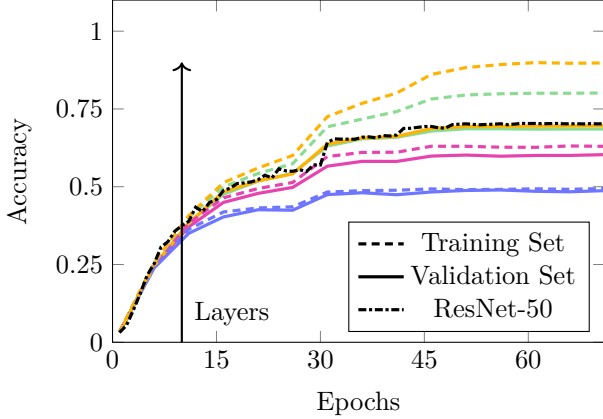

Figure 12: **Linear separability trends in ResNet-50.** Accuracy of CMs generated by LCs trained on predicted labels for layers 1 to 4, shown over the training epochs. The graph also includes the true accuracy of ResNet-50 as a baseline.

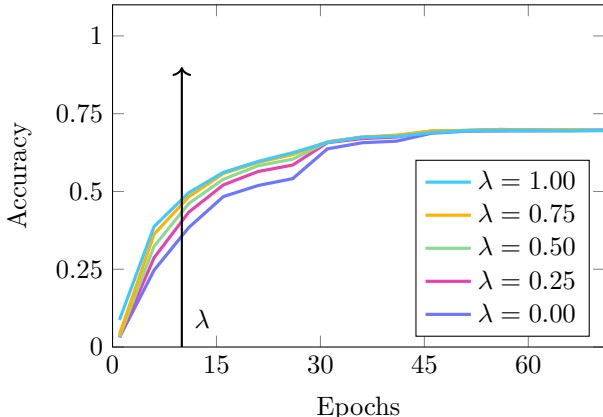

Figure 13: **Linear separability trends in ResNet-50 across several $\lambda$ values.** Accuracy of CMs generated by LCs trained on several $\lambda$ values for layer 4, shown over the training epochs for the validation set.

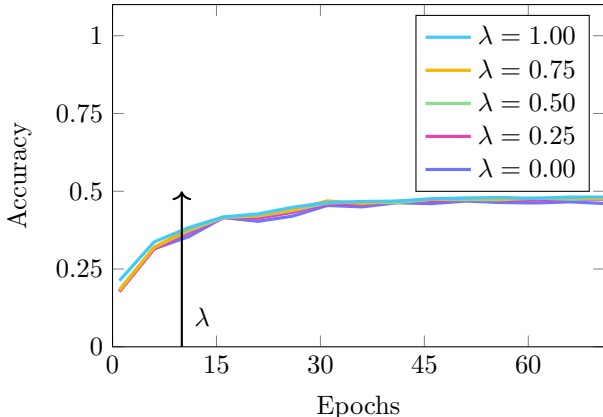

Figure 14: **Layer-wise modularity trends in ResNet-50 across several $\lambda$ values.** Modularity of CMs generated by LCs trained on several $\lambda$ values for layer 4, shown over the training epochs for the validation set.

### A.3 Custom Loss Function

As explained in Section 4.2 of the main text, the LCs are trained on a custom loss function. While we focus on the boundary cases ($\lambda = 1$ for true labels, $\lambda = 0$ for model predictions), Figures 13 and 14 show the effect of intermediate $\lambda$ values.

As discussed, training LCs on the ground truth leads to a de facto upper bound of the accuracy of the NN under analysis at this stage of training. In contrast, the accuracy of LCs trained on the model predictions can only be as accurate as the NN and thereby gives insights into the model state.

Intermediate $\lambda$ values interpolate between these two extremes, especially in the early to mid stages of the training. Once the model training converges, the curves for different $\lambda$ values also converge. Training with a mixture of true and predicted labels, such as $\lambda = 0.5$, leads to an LC that is partially informed by ground truth, while still reflecting the internal decision boundaries of the NN under analysis. As the contribution of the true labels increases, the accuracy of the LCs generally improves and the number of incorrect predictions decreases. This may make it easier to distinguish problematic confusions arising from the dataset for datasets with many classes. In regard to the modularity, we find that it is fairly stable across

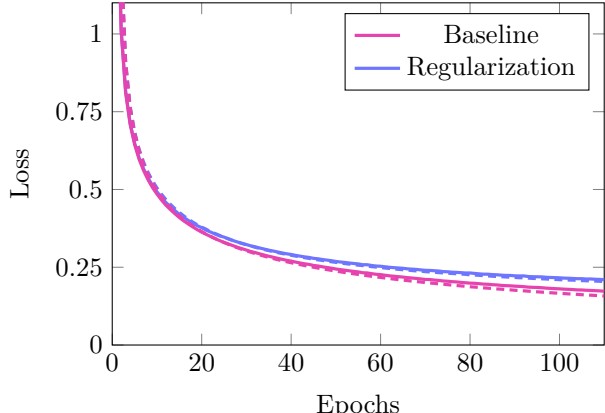

Figure 15: **Loss curves for training LCs with and without regularization.** Training (dashed) and validation (solid) loss of LCs trained on predicted labels for layer 4 of ResNet-50 at epoch 1, shown over the training epochs.

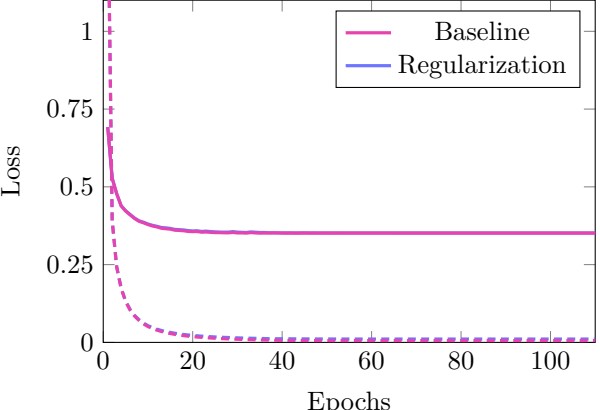

Figure 16: **Loss curves for training LCs with and without regularization.** Training (dashed) and validation (solid) loss of LCs trained on predicted labels for layer 4 of ResNet-50 at epoch 71, shown over the training epochs.

all $\lambda$ values. This suggests that meaningful group structures can be found and analyzed for any of these settings.

All $\lambda$ experiments were conducted using identical LC training settings (learning rate, batch size, optimizer, initialization), and no parameter fine-tuning was performed. The consistent behavior of accuracy and modularity across different $\lambda$ values therefore indicates that training the LCs is robust to this parameter.

### A.4 Regularization

The LCs are so far trained without regularization. To analyze the effect of weight decay (Krogh & Hertz, 1991) on the results and the LC training Figures 15 and 16 depict the loss curves of training an LC with and without weight decay for layer 4 for epochs 1 and 71, respectively. As we can see, the loss curves are very similar. This is also confirmed when looking at the accuracy and modularity of the CMs in Figures 18 and 17. As discussed in Appendix A.6 our results are robust to the LC training and this is confirmed here again.

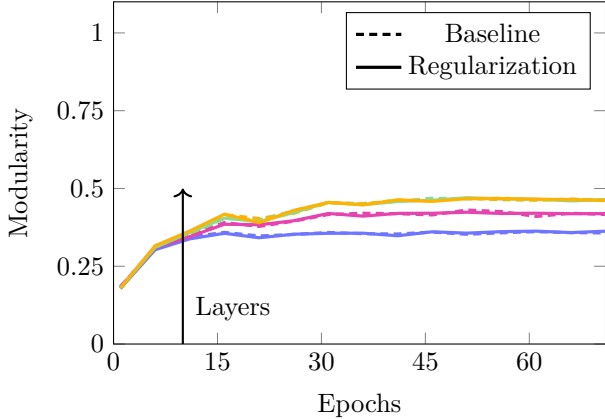

Figure 17: **Layer-wise modularity over training epochs for ResNet-50 with and without regularization.** Modularity of CCs generated from CMs for the LCs trained on predicted labels for layers 1 to 4 over the training epochs for the validation set.

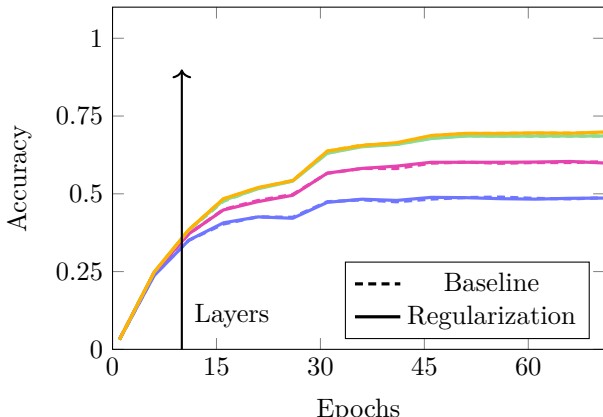

Figure 18: **Linear separability trends in ResNet-50 with and without regularization.** Accuracy of CMs generated by LCs trained on predicted labels for layers 1 to 4, shown over the training epochs for the validation set.

### A.5 Scalability

As GRAPHIC relies on visual cues to identify dataset errors, scalability to datasets with many classes is discussed here. There are several possible strategies. If visual clutter is caused by numerous confusions a fraction (e.g., 20% or 40%) can be removed for plotting only. Since all metrics can still be computed and analyzed for the dense graph, the emerging CCs are still based on the full CMs. This approach is depicted in Figure 19. Even though the number of nodes remains unchanged, the reduced edge set leads to fewer overlapping structures. This can be especially helpful in the early stages of training as there are more confusions overall.

To address larger numbers of classes, a second approach is to inspect individual CCs instead of the full graph. The strongest confusions typically cluster together, so looking at CCs one by one greatly reduces complexity while preserving the relevant structure. As an example of this, Figure 20 depicts the CCs of the animals and creepy-crawlies. As there are far fewer classes, dominant and, to human interpreters, surprising confusions can easily be spotted. This scalability approach is also not limited to just CCs in general, any subset of nodes of interest can be analyzed that way. The CCs are, however, a suitable subset as explained.

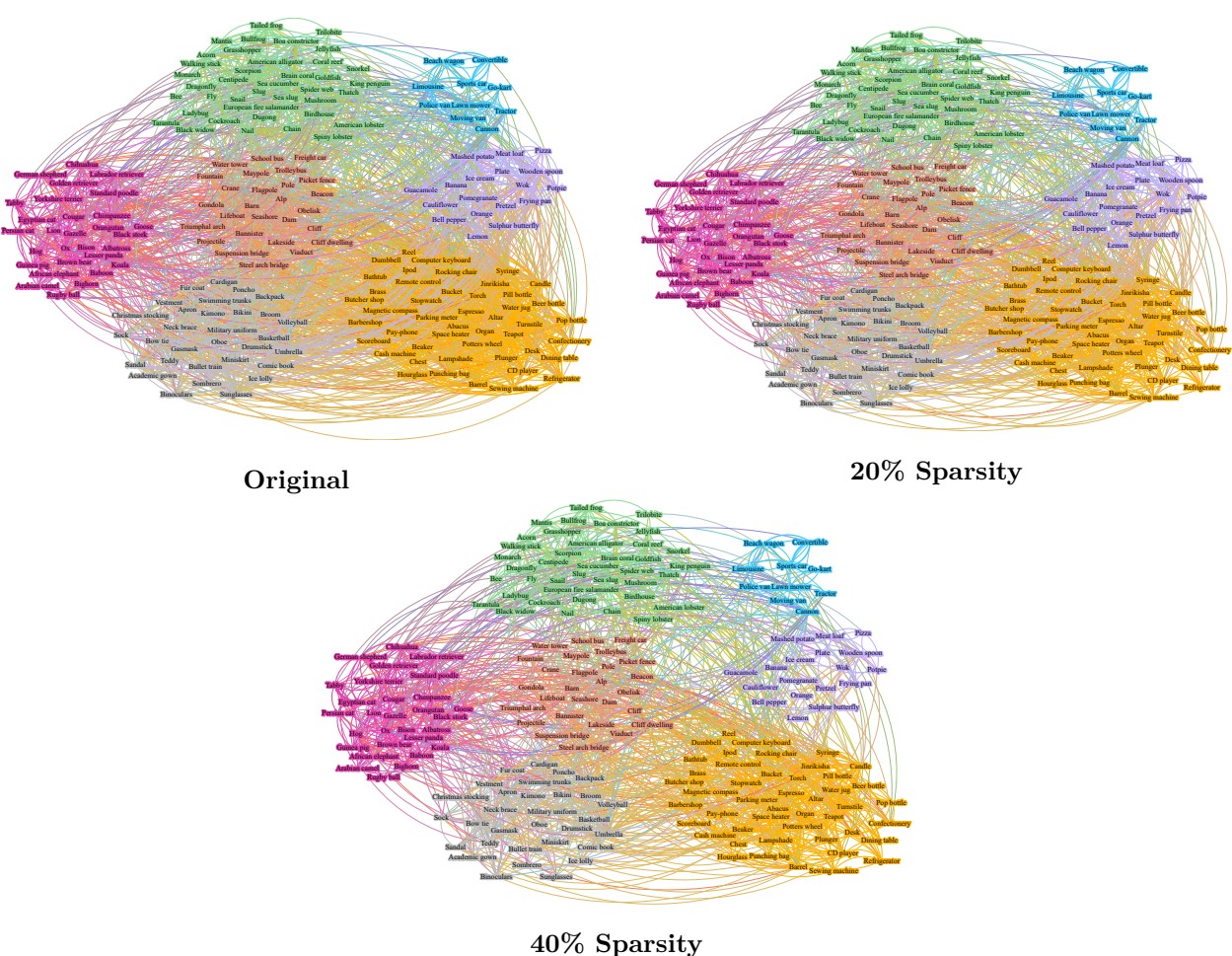

Figure 19: **Sparse confusion graphs of EffVit for Tiny ImageNet.** Visualization of CCs for decoder 12 at the final epoch for the validation set (left), the graph with 20% of the edges removed (right), and the graph with 40% of the edges removed (bottom).

As a complementary insight to understand how CCs interact with each other, nodes could also be aggregated to supernodes. A straightforward way to do that is to group all nodes in a CC together into one node representing that community, in the example, one could imagine "animals" and "creepy-crawlies" as two such supernodes. Edges between these supernodes would then be derived by summing the weights of all edges that originally connected the corresponding classes. This would provide the missing information when analyzing CCs individually.

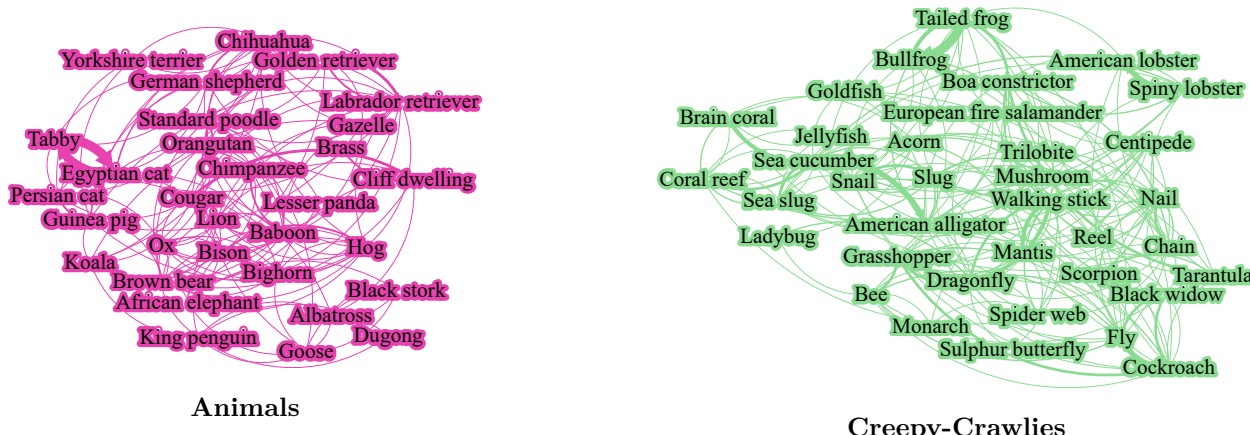

Figure 20: **Separate CC graphs of EffVit for Tiny ImageNet.** Visualization of the CCs of the animals (left) and creepy-crawlies (right) for decoder 12 at the final epoch for the validation set.

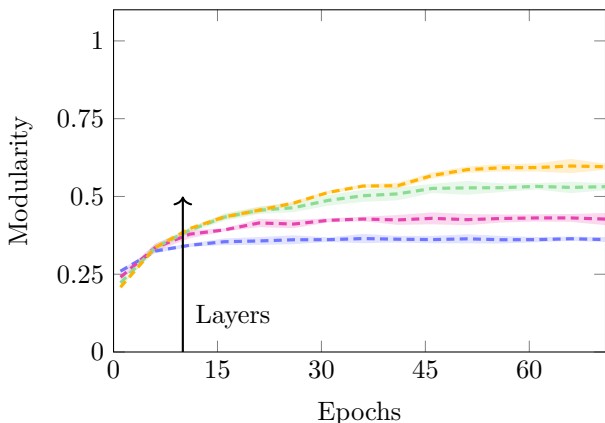

Figure 21: **Mean layer-wise modularity over training epochs for ResNet-50.** Mean modularity with three standard deviations of CCs generated from CMs for the LCs trained on true labels for the training set for layers 1 to 4 over the training epochs. Results are averaged over five seeds.

### A.6 Robustness of Linear Classifier Training

To assess the sensitivity of LCs to initialization, we train them on the same features using five different seeds. In Figure 21, we plot the mean and three standard deviations of the modularity. Both mean and its variation across seeds are very stable. While the confusion graphs for different seeds and epochs may not look identical, the main characteristics remain consistent. This confirms that the training of LCs is largely independent of initialization, and that the observed CCs are due to the underlying features rather than random factors in training. Furthermore, we analyze the robustness of the LC training to changes in the learning rates and batch sizes. For this we trained LCs on layer 4 of ResNet-50 for different batch sizes and learning rates. Figures 22 and 23 as well as Figures 24 and 25 show the loss curves of the training of layer 4 for different batch sizes and learning rates, respectively. While we see differences in the loss curves, the resulting CMs are very similar and the accuracies in Figures 26 and 27 show only negligible variations. These findings support the robustness of our approach and reinforce the reliability of the insights drawn from GRAPHIC.

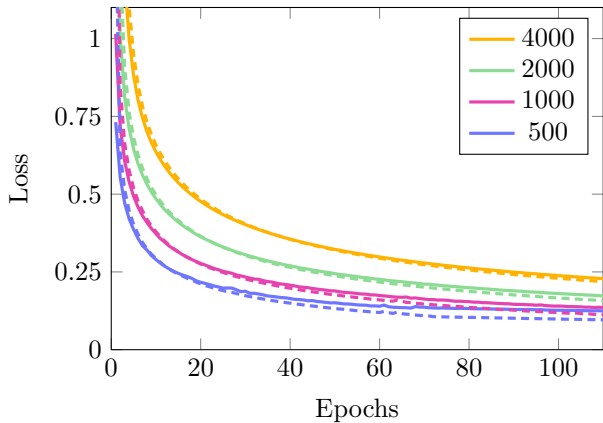

Figure 22: **Loss curves for training LCs across several batch sizes.** Training (dashed) and validation (solid) loss of LCs trained on predicted labels for layer 4 of ResNet-50 at epoch 1, shown over the training epochs.

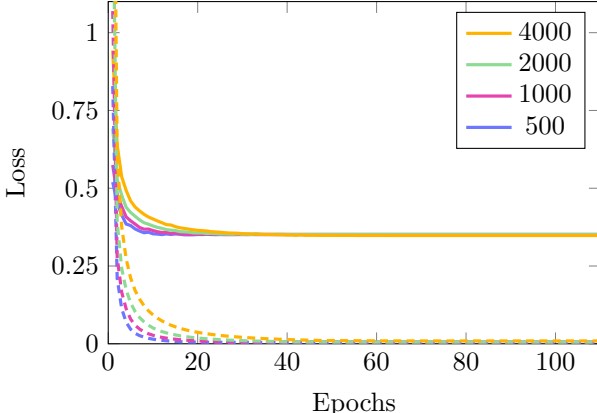

Figure 23: **Loss curves for training LCs across several batch sizes.** Training (dashed) and validation (solid) loss of LCs trained on predicted labels for layer 4 of ResNet-50 at epoch 71, shown over the training epochs.

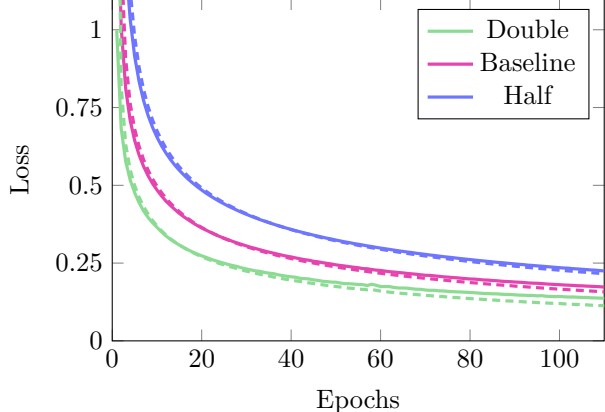

Figure 24: **Loss curves for training LCs across several learning rates.** Training (dashed) and validation (solid) loss of LCs trained on predicted labels for layer 4 of ResNet-50 at epoch 1, shown over the training epochs.

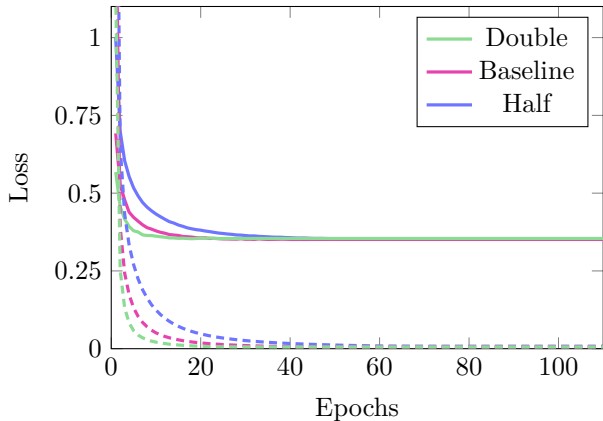

Figure 25: **Loss curves for training LCs across several learning rates.** Training (dashed) and validation (solid) loss of LCs trained on predicted labels for layer 4 of ResNet-50 at epoch 71, shown over the training epochs.

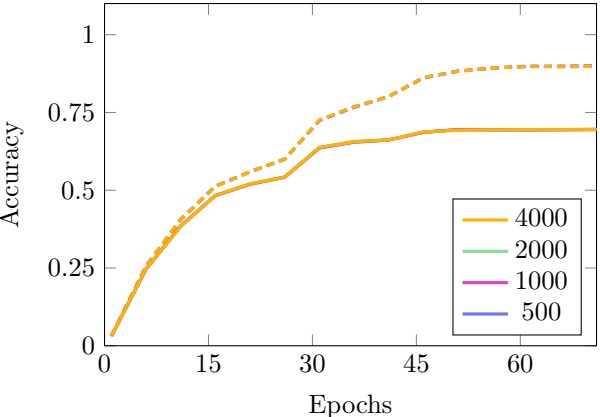

Figure 26: **Linear separability trends in ResNet-50 across several batch sizes.** Accuracy of CMs generated by LCs trained on predicted labels for layer 4, shown over the training epochs for the training (dashed) and the validation (solid) set.

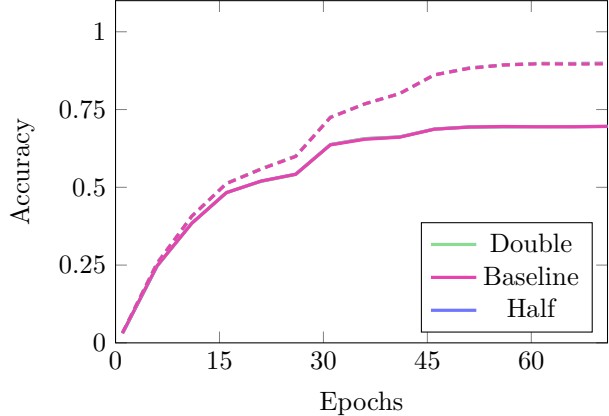

Figure 27: **Linear separability trends in ResNet-50 across several learning rates.** Accuracy of CMs generated by LCs trained on predicted labels for layer 4, shown over the training epochs for the training (dashed) and the validation (solid) set.

Table 1: **Wall-clock times for training LCs on ResNet-50 for CIFAR-100.** Total times for full training and with early stopping enabled on four different GPUs.

| GPU Model | Layer | Without Early Stopping | With Early Stopping |
|---|---|---|---|
| NVIDIA A100 | Layer 1 | 17,418s (approx. 5h) | 5,247s (approx. 1.5h) |
| | Layer 2 | 18,011s (approx. 5h) | 5,022s (approx. 1.5h) |
| | Layer 3 | 18,347s (approx. 5.5h) | 6,589s (approx. 2h) |
| | Layer 4 | 20,559s (approx. 6h) | 8,749s (approx. 2.5h) |
| NVIDIA V100 | Layer 1 | 24,136s (approx. 7h) | 7,389s (approx. 2h) |
| | Layer 2 | 25,934s (approx. 7h) | 7,377s (approx. 2h) |
| | Layer 3 | 28,513s (approx. 8h) | 9,235s (approx. 2.5h) |
| | Layer 4 | 31,121s (approx. 9h) | 13,190s (approx. 4h) |
| NVIDIA RTX 3080 | Layer 1 | 19,988s (approx. 5.5h) | 6,071s (approx. 2h) |
| | Layer 2 | 21,124s (approx. 6h) | 6,201s (approx. 2h) |
| | Layer 3 | 22,777s (approx. 6.5h) | 7,315s (approx. 2h) |
| | Layer 4 | 24,004s (approx. 7h) | 10,195s (approx. 3h) |
| NVIDIA RTX 2080 Ti | Layer 1 | 24,897s (approx. 7h) | 7,622s (approx. 2h) |
| | Layer 2 | 28,625s (approx. 8h) | 7,830s (approx. 2h) |
| | Layer 3 | 31,207s (approx. 9h) | 10,516s (approx. 3h) |
| | Layer 4 | 33,004s (approx. 9h) | 13,987s (approx. 4h) |

## A.7 Computational Overhead

Training LCs at multiple layers and epochs introduces a non-negligible computational overhead. To support practitioners in assessing what analyses are feasible for their own models and datasets, we discuss the wall-clock time required for generating CMs with GRAPHIC. All experiments in this section were conducted on ResNet-50 trained on CIFAR-100.

Table 1 reports the total wall-clock time required to train LCs for fifteen epochs for four layers, both for full training over 110 epochs and with early stopping enabled. Here, for early stopping we report the training time of the best LC with 5 epochs added as a simulated patience. Measurements are provided for four different GPUs: an 11 GB NVIDIA RTX 2080 Ti, a 10 GB NVIDIA RTX 3080, a 32 GB NVIDIA V100, and a 40 GB NVIDIA A100. The reported times include the training of the LCs, creating the CMs takes approximately 3s per matrix on all GPUs.

All LCs were trained for the full number of epochs and the best-performing LC subsequently selected. This choice was made to ensure comparability across experiments. We additionally observe systematic differences across layers, with LCs for earlier layers generally converging faster than deeper ones. In practical applications, however, using early stopping is advised.

To further understand the factors influencing runtime, we analyze how training time with early stopping varies dependent on the NN epoch and the layer for which the LCs are trained. Figure 28 shows that LC training is less expensive for later epochs, reflecting the increased separability of the representations and therefore quicker convergence time of the LCs. As shallower layers have smaller hidden dimensions, the training time for the LCs is generally shorter. Another general factor is how much data is used for the training.

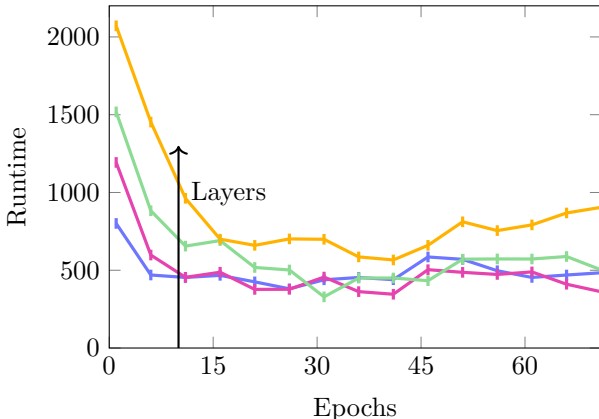

Figure 28: **Layer-wise training time of LCs over epochs.** Runtime of training the LCs on the predicted labels for layers 1 to 4, shown over the training epochs.

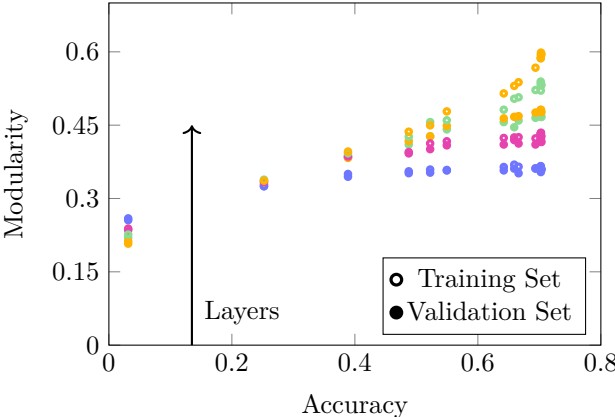

Figure 29: **Modularity accuracy trends in ResNet-50.** Modularity of CMs generated by LCs trained on true labels for layers 1 to 4, shown over the accuracy of ResNet-50.

## A.8   Practical Guidelines

To assist practitioners in choosing when and where to probe their models, we provide empirical guidelines derived from our experiments. Our analysis indicates a clear positive relationship between the accuracy of the NN and the modularity of the resulting graphs. While this relationship is not strictly linear, Figure 29 shows a consistent trend: higher accuracy generally coincides with higher modularity in all layers, with modularity eventually plateauing for earlier layers.

Practitioners interested primarily in identifying mistakes in the dataset can rely more heavily on the final epochs, where the model has converged; however, as with the flatfish man confusion, not all dataset issue may be visible in the final layer. For this reason, we recommend analyzing several layers at the last epoch.

When the goal is to study the evolution of class structures, LCs should be computed for multiple epochs. To reduce training time, the accuracy of the NN can serve as a guide for selecting which epochs to probe. Early in training, when accuracy increases rapidly, smaller gaps between epochs are recommended, whereas in the later stages, larger gaps are likely sufficient. For earlier layers, once some LCs have already been trained, the accuracy of the CMs can be monitored to decide whether changes still occur and additional LCs are helpful.

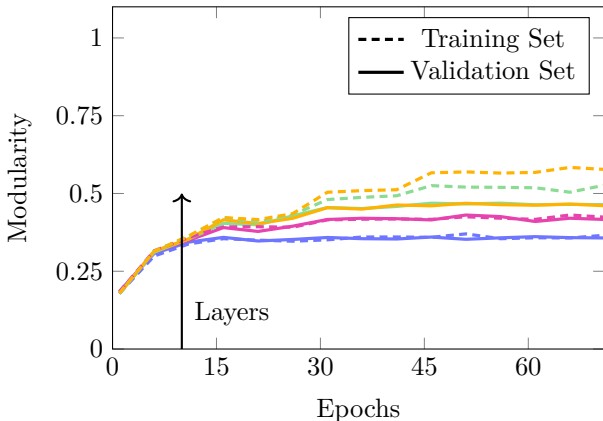

Figure 30: **Layer-wise modularity over training epochs for ResNet-50.** Modularity of CCs generated from CMs for the LCs trained on predicted labels for layers 1 to 4 over the training epochs.

### A.9 Modularity

The modularity, i.e., the measure used to group the classes and assess the strength of the grouping (cf. Section 4.1 of the main text), is plotted for both ResNet-50 and EffVit for the predicted and true labels for all layers. Modularity values below zero indicate weaker-than-random groupings, while values above 0.3 are interpreted as evidence of meaningful community structure according to Newman (2004).

For ResNet-50, the LCs trained on the predicted labels (cf. Figure 30) show an interesting trend. The modularity for layer 4 on the training set tracks the accuracy (cf. Figure 11), with similar steps. The steps stem from the used scheduler ReduceLROnPlateau (PyTorch Foundation, 2025), which reduces the learning rate if the loss stagnates. For both the predicted labels and the true labels (cf. Figure 31), the grouping strength for layers 3 and 4 is very similar.

For EffVit, the modularity of the groupings for the predicted and the true labels are depicted in Figure 32 and 33, respectively. Here, the issue mentioned earlier is also visible: for the training set with the LC trained on the predicted labels the accuracy is so high that there are almost no confusions, and the groupings become obsolete and the modularity high. This starts happening between epochs 600 and 800. For the true labels, this issue is less apparent as the accuracy is lower. The modularity may seem more volatile compared to ResNet-50, but this is just due to plotting over 1,000 epochs compared to 71.

For Tiny ImageNet, the modularity of the of CCs for CMs created with LCs trained on the true labels is illustrated in Figure 34. For later decoders, it is higher than for early ones and for the training set the modularity increases slightly starting around epoch 270. Because EffVit is trained for only 24 hours rather than to full convergence, the increase in modularity is small as the number of confusions is still relatively high.

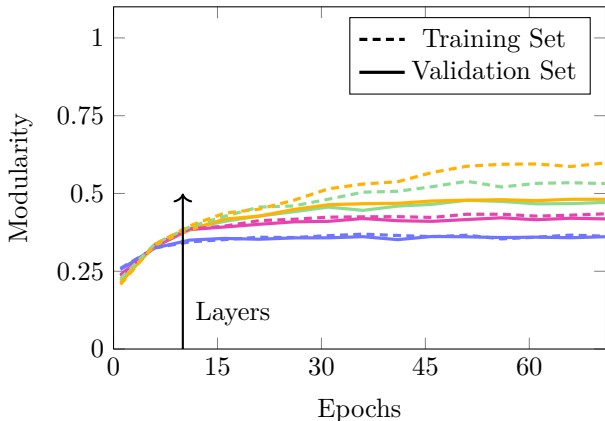

Figure 31: **Layer-wise modularity over training epochs for ResNet-50.** Modularity of CCs generated from CMs for the LCs trained on true labels for layers 1 to 4 over the training epochs.

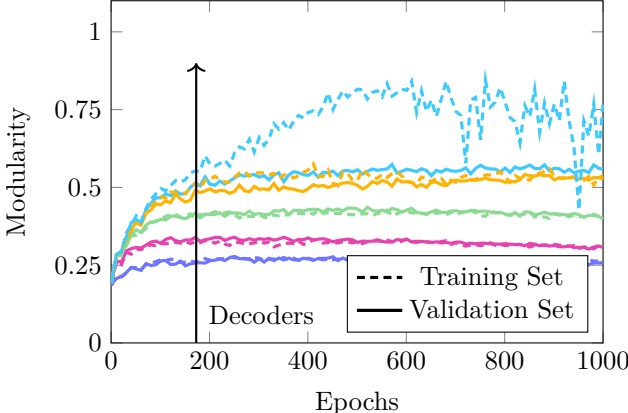

Figure 32: **Layer-wise modularity over training epochs for EffVit.** Modularity of CCs generated from CMs for the LCs trained on predicted labels for decoders 1, 3, 6, 9 and 12 over the training epochs.

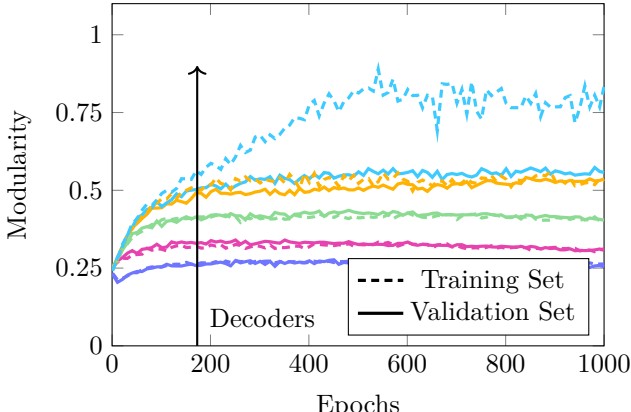

Figure 33: **Layer-wise modularity over training epochs for EffVit.** Modularity of CCs generated from CMs for the LCs trained on true labels for decoders 1, 3, 6, 9 and 12 over the training epochs.

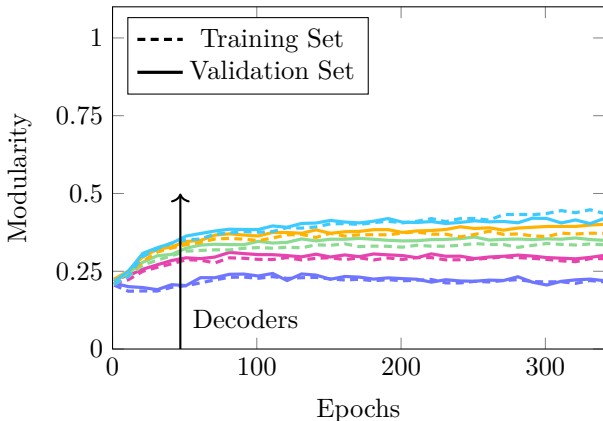

Figure 34: **Layer-wise modularity over training epochs EffVit for Tiny ImageNet.** Modularity of CCs generated from CMs for the LCs trained on true labels for decoders 1, 3, 6, 9 and 12 over the training epochs.

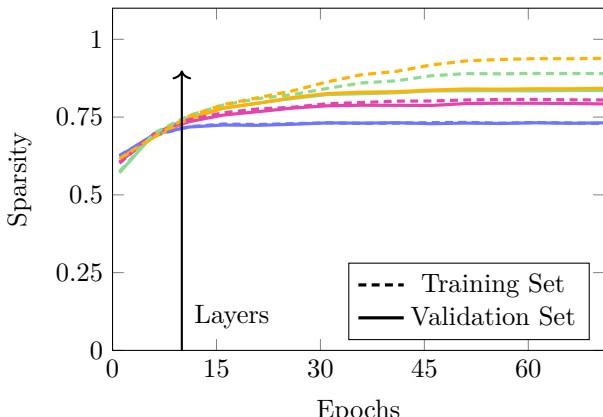

Figure 35: **Layer-wise sparsity over training epochs for ResNet-50.** Fraction of zero entries of CMs generated by LCs trained on true labels for layers 1 to 4, shown over the training epochs.

### A.10    Graph Sparsity

In addition to accuracy and modularity, we examined how sparsity evolves in the graphs over the training. The sparsity of the CMs or graphs is here defined as the percentage of zero entries of the CMs and depicted in Figures 35 and 36 for the LCs trained on the true labels and predicted labels, respectively. Here we observed an interesting development. At the beginning of training, the graphs created from LCs trained on predicted labels are considerably sparser than those created from LCs trained on the ground truth. This difference is likely caused by early layer confusion hubs: the LCs trained on predicted labels learn to reproduce these hubs, leading them to overpredict these classes while rarely predicting the remaining ones. So there are more zero entries as the in-degree of the regular classes, this is not equivalent with a high accuracy. The LCs trained on true labels do not learn to predict the confusion hubs and this sparsification does not occur.

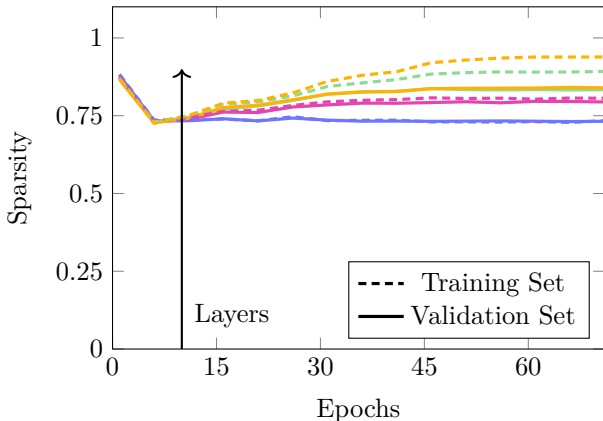

Figure 36: **Layer-wise sparsity over training epochs for ResNet-50.** Fraction of zero entries of CMs generated by LCs trained on predicted labels for layers 1 to 4, shown over the training epochs.

### A.11 Graphs

This section depicts the additional graphs created when training the LCs on the predicted labels, i.e., $\lambda = 0$, for both ResNet-50 and EffVit. Figures 37, 38, and 39 show the graphs for ResNet-50 for the validation set, for EffVit for the training and the validation set, respectively.

As mentioned in Section 5.2 of the main text, the final epoch for the training set for EffVit is less interpretable (cf. Figure 38). When the LC is trained on the predicted labels, the accuracy is so high that there are only few classes left that are incorrectly predicted, meaning that the NN cannot distinguish them. This again hints at the poor quality of the images, since, for example, leopard and tiger or worm and snake are confused.

To study how the order of training data affects the learned representations, we reversed the dataset before shuffling. Since we use a fixed random seed for shuffling, reversing the input order leads to a different shuffled sequence. Figure 40 shows the graph of layer 4 after just one epoch of training under this altered data order. Notably, compared to Figure 2 of the main text, the dominant hubs shifts from sea to road and from possum to bear, although computer keyboard remains a hub. This indicates that the early learning process – and thus the emerging structure in the graph – is partially influenced by the order in which the data is presented.

As a third aspect of the section, we show the confusion of the class flatfish with the class man. An example of the visualization of this behavior can be found in Figure 41. It depicts the human CC of layer 2 of the converged ResNet-50 model.

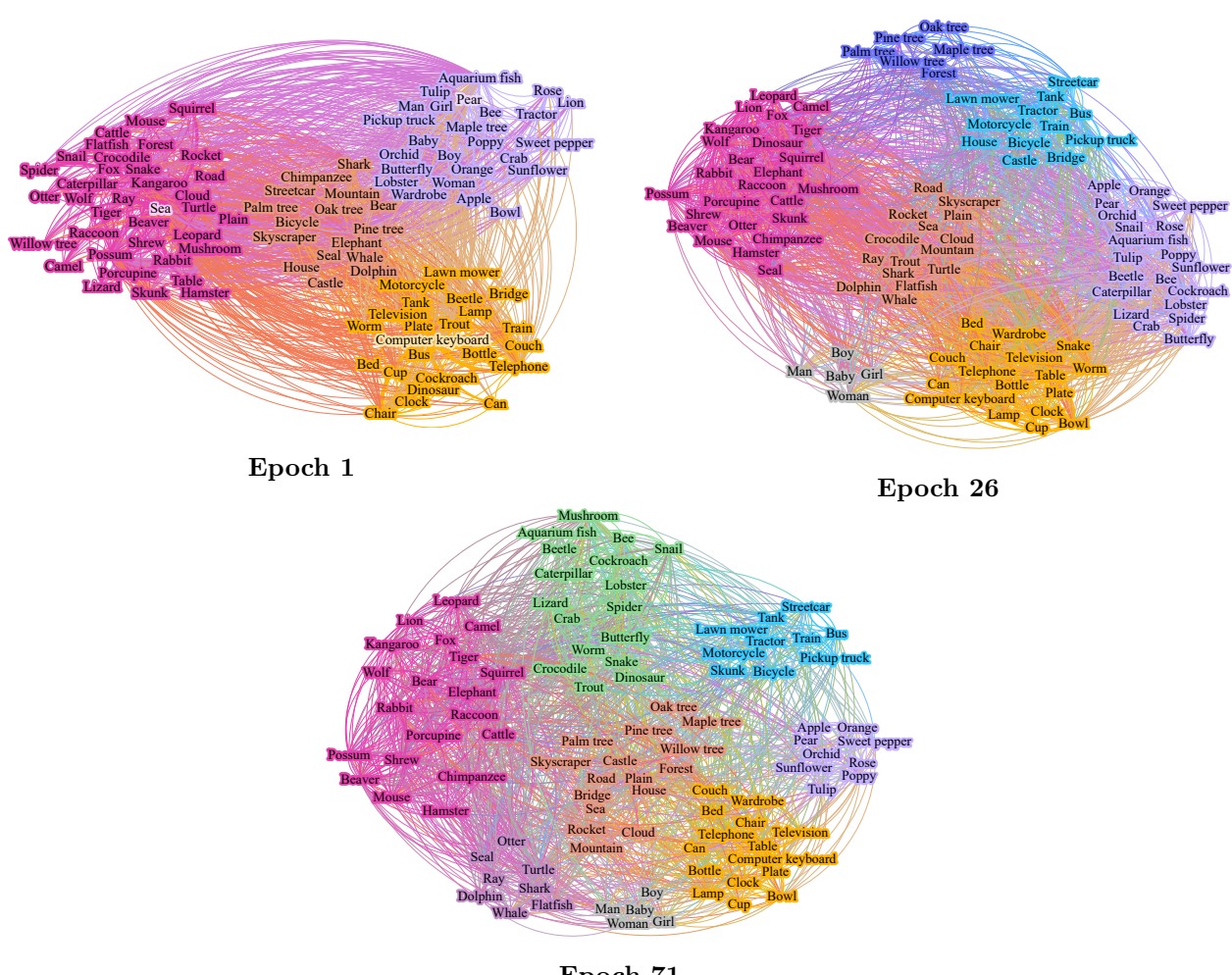

**Epoch 1**

**Epoch 26**

**Epoch 71**

Figure 37: **Confusion evolution of ResNet-50 using the validation set.** Visualization of CCs for layer 4 at early (left), intermediate (right), and final (bottom) epochs using our graph representation for the validation set.

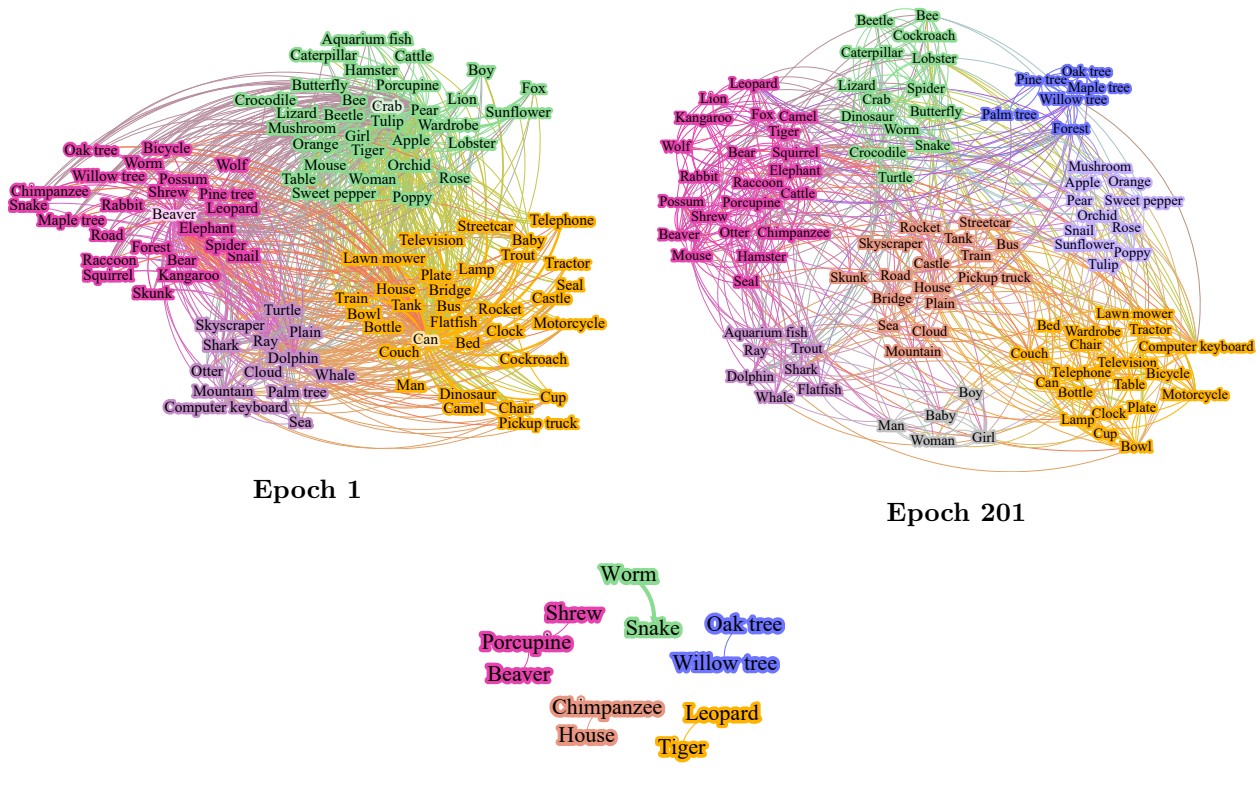

**Epoch 1**

**Epoch 201**

**Epoch 1000**

Figure 38: **Confusion evolution of EffVit using the training set.** Visualization of CCs for decoder 12 at early (left), intermediate (right), and final (bottom) epochs using our graph representation for the training set.

**Epoch 1**

**Epoch 201**

**Epoch 1000**

Figure 39: **Confusion evolution of EffVit using the validation set.** Visualization of CCs for decoder 12 at early (left), intermediate (right), and final (bottom) epochs using our graph representation for the validation set.

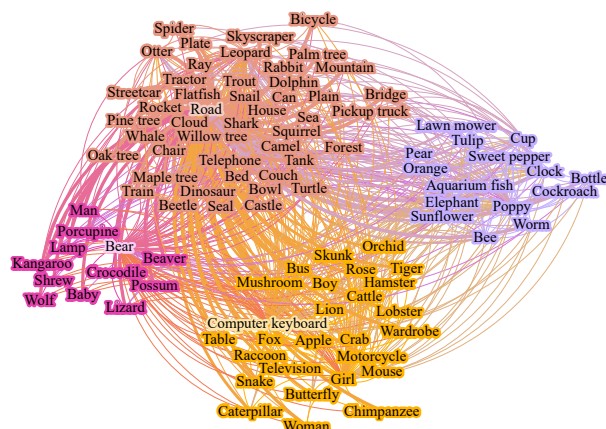

Figure 40: **Effect of dataset order on graph structure.** Graph constructed from an LC trained on the predicted labels for the training set at epoch 1 for layer 4 after reversing the dataset order.

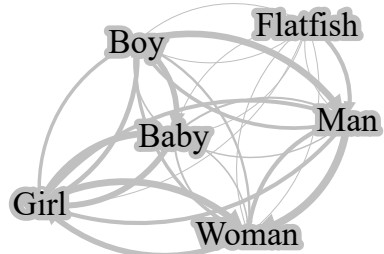

Figure 41: **CC of humans including flatfish.** Visualization of the human CC of layer 2 of ResNet-50 created using the training set including the class flatfish.

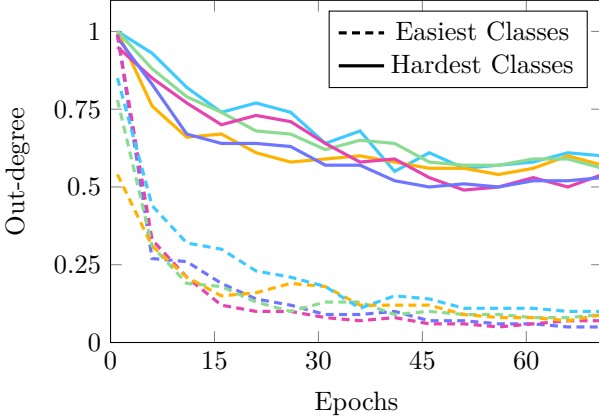

Figure 42: **Evolution of class difficulty in ResNet-50.** Out-degree of the five most difficult (solid lines) and five easiest (dashed lines) classes identified in epoch 71 for layer 4, shown over the training epochs.

## A.12   Class Difficulty

We further investigate whether certain classes are inherently difficult by analyzing the out-degree of the confusion graphs. We consider the fully converged models and identify the five classes with the highest and lowest out-degree for both ResNet-50 and EffVit. Figures 42 and 43 depict the evolution of the out-degree of these ten classes over the training epochs.

For ResNet-50, the classes with the highest out-degree are otter, man, lizard, seal, and girl (ordered from highest to lowest), while wardrobe, motorcycle, road, sunflower, and mountain are confused the least (ordered from lowest to highest). For EffVit, the most difficult classes are boy, bowl, girl, otter, and pine tree, whereas motorcycle, orange, sunflower, pickup truck, and road are among the easiest. Since the out-degree corresponds to the fraction of misclassified samples of a class and there is a clear overlap between the two models, these results suggest that some classes are intrinsically harder to recognize than others. Notably, several of the difficult classes (man, boy, and girl) align with the ambiguous labels discussed in Section 5.3.

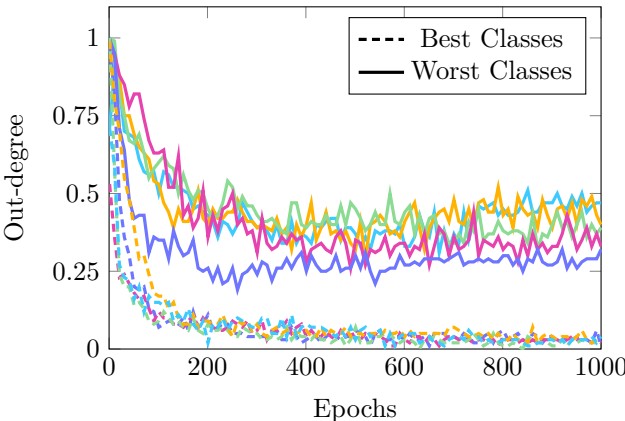

Figure 43: **Evolution of class difficulty in EffVit.** Out-degree of the five most difficult (solid lines) and five easiest (dashed lines) classes identified in epoch 1000 for decoder 12, shown over the training epochs.

Table 2: **Mapping between groups and superclasses.** Grouping of the superclasses into natural and man-made categories.

| Groups | Superclasses |
|---|---|
| Natural | aquatic mammals, fish, flowers, fruit and vegetables, insects, large carnivores, large natural outdoor scenes, large omnivores and herbivores, medium-sized mammals, non-insect invertebrates, people, reptiles, small mammals, trees |
| Man-made | food containers, household electrical device, household furniture, large man-made outdoor things, vehicles 1, vehicles 2 |

### A.13   Assortativity

We also investigate how the number of groups affects assortativity (cf. Section 5.2 of the main text). We have shown that splitting the classes into the two groups natural and man-made leads to a moderate to high assortativity for all layers. The groups are depicted in Table 2. For the evaluation of the superclasses as groups the assortativity is drastically lower. Additionally, we introduce a third grouping, where we analyze the following groups of intuitive concepts: terrestrial animals, aquatic animals, humans, everyday objects & vehicles, natural & built environments, plants & fungi. In Figure 44 we show that the assortativity for this division into six groups lies between the others, but is close to the assortativity of the two groups natural and man-made. This suggests a limited influence of the group size, but also indicates that the group size alone is not responsible for the assortativity difference.

Next, we randomly assign the classes to two sets according to the group sizes of the split into natural and man-made and compute the association matrix and with that the assortativity of the split. This is depicted in Figure 45. The assortativity is close to zero across all layers and training epochs, indicating that no genuine group structure is recovered when the assignment is random. When comparing this to splits according to the group sizes of the intuitive groups and the superclasses, a small effect of the group size is still visible. For the random grouping into two groups the assortativity is consistently the most negative, staying slightly below zero throughout training, while the assortativity of grouping into superclasses and intuitive groups at random stays around zero. The disassortative tendency only becomes noticeable in the case of two groups. Its magnitude is much smaller than the effect of a semantically meaningful grouping.

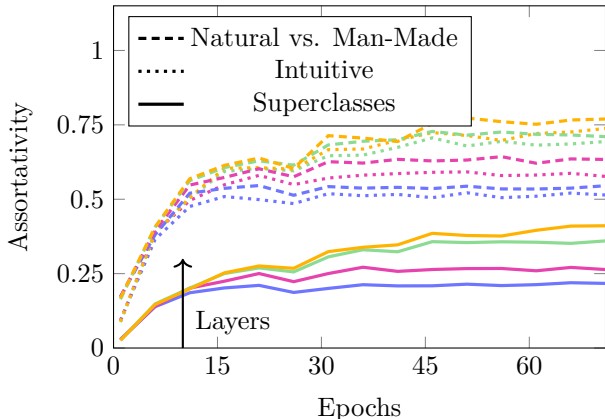

Figure 44: **Layer-wise assortativity over training epochs.** Assortativity computed by superclasses (solid lines), intuitive groups (dotted lines), and natural vs. man-made grouping (dashed lines), for layers 1 through 4 of ResNet-50.

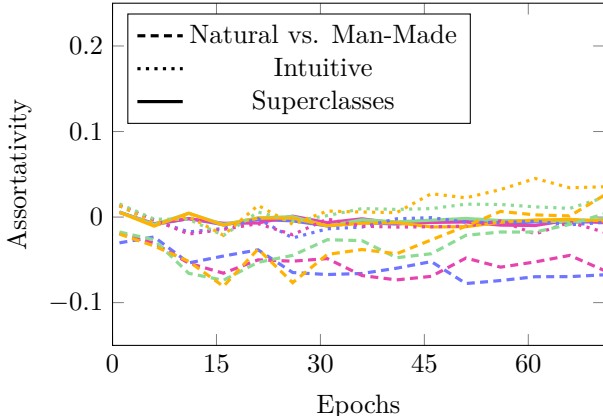

Figure 45: **Layer-wise assortativity over training epochs for random groups.** Assortativity computed for random groups with group size matching those of superclasses (solid lines), intuitive groups (dotted lines), and natural and man-made things (dashed lines), for layers 1 through 4 of ResNet-50.

## A.14    Modified Images

As previously discussed, maple trees are often depicted in fall with yellow, orange or red leaves, whereas oak trees are often depicted in green. In our representation using graphs, strong confusion between the classes becomes obvious (cf., e.g., Figure 2). To analyze the effect of the leaf color on the prediction, we change the tree color manually using the software GIMP (The GIMP Development Team, 2025). Figure 46 depicts the original and modified maple and oak trees. Three out of five images of a maple tree, originally in yellow, orange or red, were changed to green. Changing the leaf color switches two of the predictions with ResNet-50 from maple tree to oak tree. The bottom image is predicted as a willow tree instead. The top green image of a maple tree is already correctly identified as a maple tree. Here, a color change increases the probability of maple tree in the NN's output distribution from 14.62% to 15.67%. The final image is initially predicted as oak tree and through the color change, the label is corrected to maple tree. Together, these results show that there is a direct connection between the tree color and the prediction in the case of maple tree. For the oak trees the results are not as distinct. The second image from the top and the second image from the bottom were both originally predicted as oak trees and with the color change predicted as maple trees. For the three other images, the probability for maple tree increased, but the image is still predicted as oak tree.

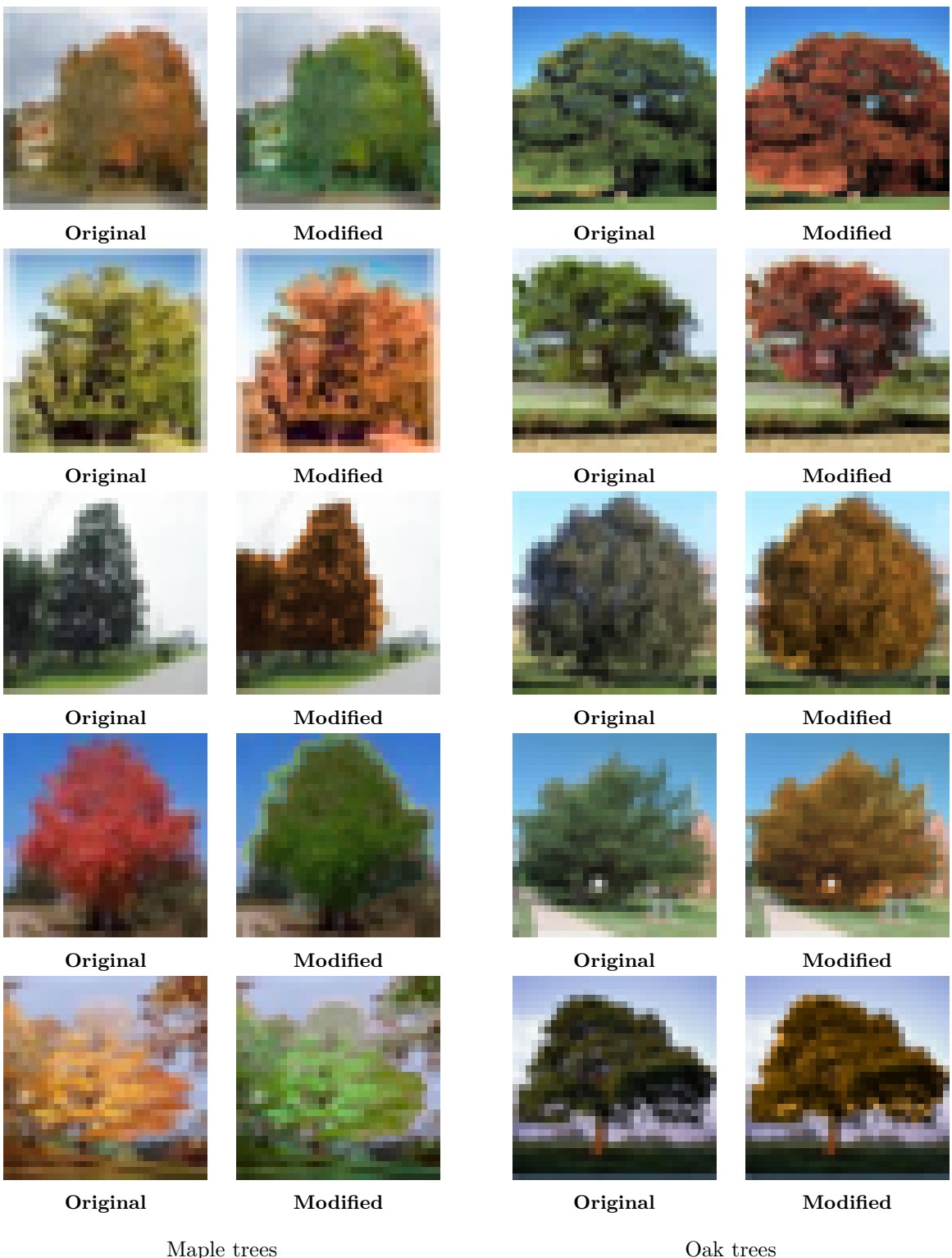

Maple trees                    Oak trees

Figure 46: **Effect of leaf color on classification for maple and oak trees.** Example images of maple and oak trees before and after color manipulation.

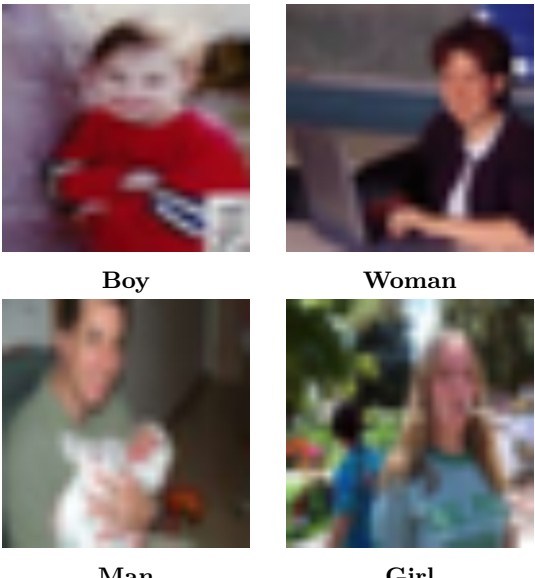

**Boy**        **Woman**

**Man**        **Girl**

Figure 47: **Images frequently misclassified by humans.** These examples are images from CIFAR-100 that were often confused by human participants during labeling. The true labels are shown below the images.

### A.15 Ambiguous Labels and Study Details

As discussed in Section 5.3 of the main text, there are images that are apparently hard to correctly label for both humans and NNs. Figure 47 depicts example images that were classified differently by many participants. Each image has the true label as its caption. The image of a boy was often confused with girl or baby as the age makes it hard to identify the gender, and there is no clear boundary between baby and toddler. The woman was confused with a man and the man was often confused with baby, as he has a baby in his arms. Finally, the girl was confused as being a woman. Again, it is not clear at what age a girl is perceived as a woman and vice versa. From the study we also derive the NN predicts women as girls and girls as women. The humans confuse girls as women, but far fewer women as girls. One reason for this may be that many images of women show them in bikinis. If the NN learns this as a main identifier, fully clothed women may be mislabeled as girls.

The study involved 31 human participants between the ages of 21 and 66. Of these, 17 identified as male and 14 as female. Seventeen participants reported using vision correction (e.g., glasses or contact lenses), while the remaining 14 did not require any correction.

### A.16 Duplicate Images

A third of the images labeled by the human participants in our study is made up of duplicate images, so images appear again in other questionnaires. We did this to evaluate the uncertainty within the labeling. Many participants split the questionnaires over several days and did not label all images at once. Even if all questionnaires were done in a single session, there are still differences in the labels of the first and second encounter. If we check the decisions per participant for each duplicate image, we find the repeat accuracy to be 83%. This means that 17% of the duplicates were labeled differently from the previous label.

One could argue that this is due to careless labeling errors – instances where participants accidentally selected a label they did not intend. Across 100 images, participants identified at most two such slip-ups in their own responses. This again suggests that the images are ambiguous. Examples of ambiguous labeling are depicted in Figure 48. For the image of the boy the accuracy for the first image is 42%, the duplicate is identified with 61% accuracy. For the image of the girl the initial labeling accuracy is 39%, the duplicate is labeled correctly with an accuracy of 23%. 71% of the participants changed their mind about their initial

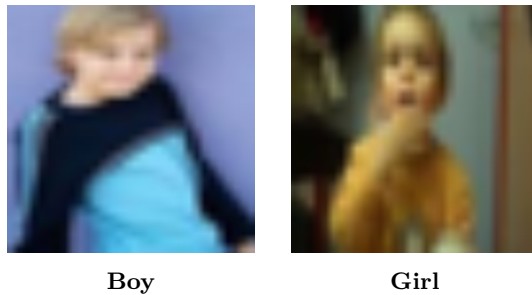

**Boy**        **Girl**

Figure 48: **Ambiguity in image labeling.** One image shows a boy (left) and was labeled as boy, girl or woman, the other image shows a girl (right) and was labeled as boy, girl, and baby by humans. For the boy 71% of participants changed their label, for the duplicate image for the girl 48%.

label of the boy, and 48% changed their label of the girl. These results are not surprising, as the age limit for baby is not clearly defined, and young boys and girls cannot be distinguished given the poor quality of the images.

