# OpenReview forum: "The Confusion is Real: GRAPHIC - A Network Science Approach to Confusion Matrices in Deep Learning"
_TMLR — Accepted by TMLR_

### Review · Reviewer_Dp2K · 2025-12-09

**Summary Of Contributions:**

The paper introduces a method to study explainability in neural networks (NNs). The authors show examples of using the proposed methodology on two vision model architectures: EffVit and ResNet-50.

**Strengths:**
1. Novel methodology that can help understand the dynamics of learning different (groups of) classes over the training epochs.
2. Very well written paper with interesting details in the appendix. The clarity of communication, eg. explaining how the graph evolved over epochs (and potentially why) is commendable.
3. The authors have understood the problem and experiment observations in depth, and have highlighted specific examples like effects of leaf color on classification, contextual bias in flatfish images (in line with previous literature in the field), and potential pitfalls at the dataset level


**Weaknesses:**
My main concern is with respect to scalability. Scalability in this case can be thought of from a few dimensions - model architecture, number of classes, computational resources.
1. Model architecture: It's great to already see two different architectures being discussed in the paper. However, some experiments/discussions about scalability to, for example, non-vision models like those meant for text classification or graph classification would help validate the generalizability more
2. Number of classes: Is the method expected to scale for, say 1000 classes? The primary way of understanding seems to be manual visual inspection - it is already very helpful, however it will face some scalability issues. One way could be proposing certain metrics, that can be tracked every 5-10 epochs, and certain inflection points might indicate considerable changes in graph structure (just a theory, needs to be tested)
3. Computational resources: Training a linear classifier for every (or a subset of) layer after every 5-10 epochs can add a computational overhead depending on certain factors like number of layers, amount of data etc. Some stats and figures about this added cost would help potential users weigh their options


Apart from the above mentioned added analysis, and stress testing, this paper is already is a great shape!

**Audience:**

Yes

**Audience Explanation:**

This would be very useful to the researchers working in explainable AI.
For those working in network science, this can open doors towards more interesting applications of graph methods in classic NNs.

**Claims And Evidence:**

Yes

**Claims Explanation:**

The experiments and in-depth analysis supports the proposed methodology

**Requested Changes:**

(Referring to a few things mentioned in the summary)

1. Experiments with other architectures: Nice to have but not necessary
2. Discussion about scalability with respect to number of classes: Critical for acceptance
3. Computational complexity and cost: Critical for acceptance

---

> ### Author Response · Authors · 2025-12-17
> **Response to Reviewer Dp2K (part 1)**
>
> We sincerely thank the reviewer for their thorough and thoughtful feedback, as well as for their strong support of our work. We appreciate the opportunity to clarify and strengthen the paper.
>
> Below, we address the reviewer’s comments point by point:
>
> > Model architecture: It's great to already see two different architectures being discussed in the paper. However, some experiments/discussions about scalability to, for example, non-vision models like those meant for text classification or graph classification would help validate the generalizability more
>
> We agree that extending the proposed methodology beyond vision models is an important direction. While the current submission focuses on image recognition tasks for vision transformers and CNNs, the methodology itself is architecture-agnostic and relies only on intermediate representations, availability of labeled data and classification tasks.
>
> That said, we acknowledge that experiments on non-vision domains (e.g., text or speech classification) would further strengthen the generality of the method. **We now explicitly discuss this in the revised conclusion**, highlighting text sentiment analysis and speech classification as promising next steps.
>
>
> > Number of classes: Is the method expected to scale for, say 1000 classes? The primary way of understanding seems to be manual visual inspection - it is already very helpful, however it will face some scalability issues. One way could be proposing certain metrics, that can be tracked every 5-10 epochs, and certain inflection points might indicate considerable changes in graph structure (just a theory, needs to be tested)
>
> We thank the reviewer for highlighting this important concern. We fully agree that visual inspection of confusion graphs becomes challenging as the number of classes increases. To address this, **we added Appendix A.5 discussing three strategies for scalability of our method** to datasets with more classes. We also argue that the model accuracy is a good indicator of where changes in the graph structure happen and discuss this **in Appendix A.8** as **advice for practitioners**.
>
> First, we evaluated the effect of **filtering confusion matrices** by removing a fixed fraction (e.g., 20% or 40%) of the smallest non-zero confusion values. This reduces the number of confusions and visual clutter as is depicted **in Appendix A.5** of the revised script. As we want to preserve the most information possible, we only removed these weights for the graph plot, but not for the computation of the metrics. This still preserves the creation of CCs and allows issues, like the man flatfish confusion, to become visible.
>
> Second, we propose to plot CCs of large graphs individually. While this does not highlight how CCs interact with each other, the strongest confusions and, as shown in our paper, dataset issues end up in the same CC. We also show **examples** of that **in Appendix A.5** of the revised script.
>
> Finally, to complement insights gained from visualizing individual CCs and gain a high level understanding of how CCs, and thereby classes, interact with each other, nodes within CCs could be aggregated into supernodes. This creates a superclass-level graph, which captures relationships between groups of classes while reducing complexity.
>
> **We added a detailed discussion and illustrative examples on scalability of our approach in Appendix A.5 of the revised paper.**
>
> > Computational resources: Training a linear classifier for every (or a subset of) layer after every 5-10 epochs can add a computational overhead depending on certain factors like number of layers, amount of data etc. Some stats and figures about this added cost would help potential users weigh their options
>
> We recognize that a detailed **breakdown of the computational resources** and time demand for our experiments will help practitioners to decide what is doable for them and what they can analyze in their datasets or models. This is now discussed **in Appendix A.7**. We, therefore, provide wall-clock times of creating our CMs on four different GPUs, namely 11 GB NVIDIA RTX 2080 Ti, 10 GB NVIDIA RTX 3080, 32 GB NVIDIA V100, and 40 GB NVIDIA A100 GPU. In the table below the wall-clock time for the training of the LCs on ResNet-50 and CIFAR-100 for both full training (110 epochs) and with early stopping. For early stopping we report the training time of the best LC with 5 epochs added as a simulated patience. For research purposes, we have used full training and then selected the best-performing classifier; however, early stopping will be the default for the final code to reduce runtime for practitioners.

---

> > ### Author Response · Authors · 2025-12-17
> > **Response to Reviewer Dp2K (part 2)**
> >
> > | GPU Model       | Without Early Stopping  | With Early Stopping|
> > |-----------------|----------------------------|------------------------|
> > | NVIDIA A100     |             21.5h               |          7.5h              |
> > | NVIDIA V100     |                31h           |              10.5h          |
> > | NVIDIA RTX 3080 |              25h              |             9h           |
> > | NVIDIA RTX 2080 Ti |            33h             |              11h          |
> >
> >
> > We also provide insights into the influence factors of the training time. For that, we plot the training time with early stopping over the epochs, for which the LCs were trained for. We also report how this changes dependent on the layer. And, as mentioned above, give **practical guidelines** on which layers and epochs are most relevant for a resource efficient analysis **in Appendix A.8**.
> >
> > **In summary, we have added Appendix A.7 to the revised script to give a comprehensive overview of the computational complexity of our approach.**

---

### Review · Reviewer_pw9o · 2025-12-09

**Summary Of Contributions:**

This paper introduces GRAPHIC, a global, class-level explainability framework that constructs confusion matrices at intermediate layers of a neural network by training linear classifiers (LCs) on hidden-layer representations. It treats these confusion matrices as weighted directed graphs, enabling analysis using network science tools. It provides a systematic, architecture-agnostic way to visualize and interpret neural training dynamics on a class level, showing architectural differences in learning, particularly in linear separability.

**Audience:**

Yes

**Audience Explanation:**

Yes, GRAPHIC sits at the intersection of explainability, representation learning, and dataset diagnostics. Specifically, this method is architecture-agnostic, appealing to both CNN and transformer researchers. Also, the separability analysis for ViTs addresses an open question in how transformers learn vs. CNNs.

**Broader Impact Concerns:**

No concerns about the ethical implications of this work

**Claims And Evidence:**

Yes

**Claims Explanation:**

1. Qualitative graphs are rich, but the claims would benefit from more numerical summaries, e.g., average community modularity trends, measures of graph sparsity, confusion entropy, etc. Currently, most conclusions rely on visual inspection.
2. Robustness to LC training choices is not deeply explored. Appendix A.3 asserts robustness to initialization, but not to (1) LC architecture variations and (2) different $\lambda$ values.

**Requested Changes:**

1. It would be helpful to add quantitative metrics to complement qualitative graph visualizations, such as summaries of modularity over epochs, numeric instability measures of confusions, graph sparsification statistics, mutual information between communities and human-defined groups
2. It would be nice to clarify LC training robustness, such as demonstrating stability across LC learning rates / batch sizes, comparing LC vs simple nearest-centroid or cosine-classifiers to ensure conclusions are not LC-specific, etc.
3. Please provide complexity analysis and practical guidance. The author note that running LC training at every 5–10 epochs can be expensive, so a more systematic cost breakdown would be helpful; It also helps to suggest heuristics for selecting which layers to probe.
4. Comparisons to existing global XAI: How does GRAPHIC compare to t-SNE/UMAP visualization for detecting dataset artifacts? To what extent does GRAPHIC overlap with ConfusionFlow? How does class-level probing differ from TCAV or concept bottleneck models?

---

> ### Author Response · Authors · 2025-12-17
> **Response to Reviewer pw9o**
>
> We thank the reviewer for their detailed and constructive feedback, as well as for recognizing the relevance and broad applicability of our method.
>
> Below, we address the reviewer's comments point by point:
>
> >Robustness to LC training choices is not deeply explored. Appendix A.3 asserts robustness to initialization, but not to (1) LC architecture variations and (2) different $\lambda$  values.
>
> We thank the reviewer for pointing out the need to examine robustness with respect to the $\lambda$ parameter. **Appendix A.3** discusses the **custom loss function** and the resulting linear separability trends. All experiments across different $\lambda$ values use identical LC training settings (same learning rate, batch size, and optimizer), ensuring that $\lambda$ is the only varying factor. These results did not require any hyperparameter fine-tuning, which suggests that the LC training is robust to these variations. We have also added a plot on how the modularity varies for different $\lambda$ values to support this observation.
>
> **We are now explicitly discussing this in Appendix A.3 to show that this parameter does not influence the robustness of our LCs.**
>
> > It would be nice to clarify LC training robustness, such as demonstrating stability across LC learning rates / batch sizes, comparing LC vs simple nearest-centroid or cosine-classifiers to ensure conclusions are not LC-specific, etc.
>
> We agree that our work would benefit from more details on the **robustness of the LC training** and how different choices for learning rates and batch sizes influence the stability of the training and our results. To address this, we have added experiments to **Appendix A.6**, where we show exemplary loss curves of training the LCs for different learning rates and batch sizes as well as plots on how this affects the accuracy. We can see that while the loss curves vary slightly, the accuracy of the CMs is not affected much.
>
>
> We are open to extending our approach with other classifiers in the future, as this may lead to shorter training times, which benefits the practitioner.
>
> **The additional experiments suggest that our conclusions are not sensitive to moderate changes in LC training hyperparameters and are found in Appendix A.6**.
>
> > It would be helpful to add quantitative metrics to complement qualitative graph visualizations, such as summaries of modularity over epochs, numeric instability measures of confusions, graph sparsification statistics, mutual information between communities and human-defined groups
>
> We thank the reviewer for encouraging us to extend the quantitative analysis. In **Appendix A.9** we already discuss the modularity evolution over the epochs, but have now added **Appendix A.10** to analyze the **evolution of the sparsity** of the graphs. Here, we found an interesting development. Initially, the graphs created from the LCs trained on the predicted labels are sparser than the ones created with the ground truth. This effect is likely caused by the early layer confusion hubs. The LCs trained on the predicted labels basically learn to predict the confusion hubs, while the LCs trained on the true labels would learn to not predict the confusion hubs. This is also reflected by the difference in accuracy described in **Appendix A.2**.
>
> **We have added Appendix A.10 to our paper to discuss graph sparsity over the training.**
>
> > Please provide complexity analysis and practical guidance. The author note that running LC training at every 5–10 epochs can be expensive, so a more systematic cost breakdown would be helpful; It also helps to suggest heuristics for selecting which layers to probe.
>
> We appreciate the reviewer's feedback and have added **Appendices A.7 and A.8** to discuss the **wall-clock time and aid practitioners in running their experiments**. We provide timing results on 4 different GPUs. In terms of practical advice for the practitioners, we suggest the following guidelines:
> We generally advise using the accuracy of the NN to assess where larger changes in the graphs happen. We show in **Appendix A.8** that the modularity, so the strength of the grouping, is related to the accuracy of the NN. Higher accuracy generally coincides with higher modularity, even though the modularity for early layer stagnates at some point.

---

> > ### Author Response · Authors · 2025-12-17
> > **Response to Reviewer pw9o (part 2)**
> >
> > We identify two strategies based on the desired outcome. If practitioners are interested in analyzing their dataset for mistakes, the layers of the fully trained network are most relevant. To analyze the training process, we recommend the following strategy for layer and epoch selection. We suggest identifying stages of training based on the accuracy. Early in training, when accuracy increases quickly, it can be useful to sample more frequently to capture rapid changes in class structures. Later in training, once accuracy has plateaued, fewer checkpoints are likely sufficient. For earlier layers, monitoring the accuracy of the trained LCs can indicate when further training provides little additional insight, allowing one to stop training those LCs early and save computational resources.
> >
> > **In summary, we have added Appendix A.7 and A.8 discussing layer and epoch selection strategies as well as wall-clock training times.**
> >
> > > Comparisons to existing global XAI: How does GRAPHIC compare to t-SNE/UMAP visualization for detecting dataset artifacts? To what extent does GRAPHIC overlap with ConfusionFlow? How does class-level probing differ from TCAV or concept bottleneck models?
> >
> > We agree that this is not fully discussed in our paper yet and have added **Appendix A.1** to clarify how GRAPHIC **relates to existing explainability approaches**.
> >
> > To summarize some findings, t-SNE and UMAP visualize datasets by mapping samples from their high dimensional space to a low dimensional space. While this can lead to conclusions about classes, this is not the main focus.
> >
> > ConfusionFlow visualizes class confusions over time directly in the confusion matrix, but is limited to small datasets. Our approach with GRAPHIC represents confusions as weighted graphs, scales to large numbers of classes, and leverages network science to analyze class relations across layers.
> >
> > TCAV and concept bottleneck models require datasets with concept-level annotations. For example, the fictional classes “zebra” and “crosswalk” both contain the concept “stripes,” whereas “horse” may be visually similar to “zebra” but does not share this concept. In contrast, our analysis operates at the class level, distinguishing individual classes rather than shared concepts. If a dataset with concept annotations were available, GRAPHIC could in principle be extended to visualize concept-level relationships.
> >
> >
> > **In summary, we have added Appendix A.1 to the revised manuscript to compare GRAPHIC to other methods.**

---

### Review · Reviewer_uBCv · 2025-12-09

**Summary Of Contributions:**

The work proposes an approach to investigate neural networks
by training logistic regression on the activation embedding
of each layer and analyzing the resulting confusion matrices.
The logistic regression is trained not only on the labels,
but on a convex combination of the loss on the labels,
and the prediction of the full neural network.
It provides a demonstrations of the efficacy of the approach
by identifying dataset issues on CIFAR-100,
as well as some observations of the training behavior.


### Strengths

1. The human study to verify the Baby/Man/Woman/Girl/Boy cluster is valuable
   evidence to back the identified confusion in ResNet50.
2. The overall approach is straight forward, being a direct combinations of two prior works (Alain et al., 2017; Jin et al., 2017).
3. The approach gives insight into the training dynamics of the model,
   something that is fully ignored by post-hoc methods.
4. The approach is tested on both CNNs and transformer models.

5. The writing of the work is very good, including its sectioning and individual paragaphs.
6. The "Main Takeaway" paragraphs are very helpful in quickly understanding the results.

### Weaknesses

1. The work does not directly compare to previous works. Specifically, it is
   implied that looking only at the top-$\tau$ predicted classes, and only
   looking at the final layer of the converged model in Jin et al. (2017) misses
   important details, but a specific comparison is missing.
2. A limitation of the method, and also of its evaluation,
   is that it requires a manual analysis of a human expert who understands the
   semantic connections of the analysed classes. The claim for its efficacy is
   only backed by observations of the authors.
3. The approach heavily depends on the classes and their semantic connections.
   It will not provide any insight in settings with a low number of classes (e.g., binary classification).
4. Although the training-time analysis is interesting,
   especially with respect to the fact that the order appears to matter,
   there does not seem to be any actionable insight specifically for this analysis.

**Audience:**

Yes

**Audience Explanation:**

The work provides a clear extension of two prior works (Alain et al., 2017; Jin et al., 2017),
the combination of which (i.e., conducting the analysis of Jin et al. (2017) on all layers)
appears very relevant and insightful.
Furthermore, the work analyzes the change of the model confusions over the training epochs,
which is an important detail many post-hoc approaches ignore.
The observed model behavior may potentially aid in curriculum learning,
which is a very important and fundamental problem in machine learning,
and thus of interest to TMLR's audience.

**Broader Impact Concerns:**

I do not see any ethical implications of this work that would require a broader
impact statement.

**Claims And Evidence:**

Yes

**Claims Explanation:**

While the work lacks direct evidence of the general efficacy of the proposed method,
evidence that could only be provided through a thorough human-study,
it does not directly claim efficacy. Rather, it only claims to demonstrate
specific cases in which some understanding of the neral network training can
be obtained.

**Requested Changes:**

1. (critical) I think it is critical to directly compare what insights could be made by
   only looking at the final layer of the converged model as in Jin et al. (2017).
   It is likely that all the dataset issues in 5.3 can already be made by the
   approach proposed by Jin et al. (2017), which would not require the more
   extensive analysis of the proposed method.

2. (stronger) As the order seems to matter, could this be used for curriculum learning?
  A discussion and respective experiment would provide actionable insight,
  and signficiantly strengthen the work.

3. (stronger) I am unsure how insightful the linear separability study is. Previous work
   has discussed that certain classes are more separable in certain layers, but
   I do not see how actionable that insight is.

4. (stronger) I did not fully understand the purpose of the distillation loss.
   It will cause some of the geometry learned in higher layers to be embedded
   into the lower layers, so the resulting linear classifiers may not be a good
   representation of the learned embedding up to that layer.

5. (stronger) The loss in (4) does not seem to include a regularization term,
   which might cause some issues related to the uniqueness of the solution of the linear classifier.

### Minor

- groups have index c, vertices have index i. This could be confusing, given that classes are usually index with c

- In (2): maybe write $\sum_{i\in\mathbb{G}_u} \sum_{j\in\mathbb{G}_v} C^\text{ad}_{i,j}$
- in (1), the adjacency matrix is using the wrong symbol, it should be bold
    - if you meant to use the non-bold version to identify as a scalar, it conflicts with the number of classes C

- (Alain et al., 2017) seems to omit Yoshua Bengio in the reference

- distillation (Hinton et al., 2015) could be cited for the distillation loss

---

> ### Author Response · Authors · 2025-12-17
> **Response to Reviewer uBCv (part 1)**
>
> We appreciate the reviewer's careful and well-reasoned assessment of our work and are glad for the opportunity to refine and improve our manuscript.
>
> Below, we address the reviewer's comments point by point:
>
> > The work does not directly compare to previous works. Specifically, it is implied that looking only at the top-$\tau$ predicted classes, and only looking at the final layer of the converged model in Jin et al. (2017) misses important details, but a specific comparison is missing.
>
> > (critical) I think it is critical to directly compare what insights could be made by only looking at the final layer of the converged model as in Jin et al. (2017). It is likely that all the dataset issues in 5.3 can already be made by the approach proposed by Jin et al. (2017), which would not require the more extensive analysis of the proposed method.
>
> We agree that our work so far does not address a direct comparison between previous work and our method and that this will be helpful to readers. We have, therefore, added **Appendix A.1** to **provide a clear comparison between previous work and our method**. We found that looking at the top-$\tau$ predictions in the work by Jin et al. (2017) misses, e.g., the connection between the classes man and flatfish. While this is also not a prominent connection in the final converged layer for our graphs, our representation consistently finds this confusion. This is, however, also why we suggest to use not only the final layer, but earlier layers as well. The man flatfish confusion is easily spotted in the converged layer 2 as flatfish is part of the CC of the humans, but hard to detect in the final layer.
>
> Another key difference is the utilization of directed instead of undirected graphs. Jin et al. (2017) suggest that trees are confused due to similar color and texture. We generally agree with this assessment, but additionally found, that the specific confusion between maple tree and oak tree is due to seasonal cues. In our representation especially the confusion of maple trees as oak trees stands out. The connection is reciprocal, but oak trees are less often confused as maple trees. If unweighted edges are used this confusion becomes averaged and is less prominent in comparison to the other tree confusions.
>
> Finally, GRAPHIC extends the analysis beyond the converged NN by explicitly incorporating temporal dynamics, enabling the study of how confusions form and evolve during training.
>
> **In summary, we have added Appendix A.1 to discuss similarities and differences between our approach and previous work.**
>
> > A limitation of the method, and also of its evaluation, is that it requires a manual analysis of a human expert who understands the semantic connections of the analysed classes. The claim for its efficacy is only backed by observations of the authors.
>
> We agree that **our methods requires manual evaluations** to reach full potential. While general trends of the NN training like the linear separability per layer or the increase in modularity do not require manual intervention, interpreting changes semantically meaningful is based on human evalutions. **We now highlight this limitation in the conclusion.**
>
> > The approach heavily depends on the classes and their semantic connections. It will not provide any insight in settings with a low number of classes (e.g., binary classification).
>
> We appreciate this comment and agree that in settings with very few classes, such as binary classification, the benefits of analyzing class-level confusions are very limited, as there is little structure to uncover. In such cases, GRAPHIC could be adapted to **operate on concepts** or additional labels representing semantically relevant features, effectively performing a concept-level analysis. This allows a similar analysis even when the number of classes is small. The main drawback of this is the need of additional labels. We also discuss this extension when comparing to other concept-based explainability methods in **Appendix A.1**.
>
> > Although the training-time analysis is interesting, especially with respect to the fact that the order appears to matter, there does not seem to be any actionable insight specifically for this analysis.
>
> > (stronger) As the order seems to matter, could this be used for curriculum learning? A discussion and respective experiment would provide actionable insight, and signficiantly strengthen the work.
>
> We thank the reviewer for pointing out this concern and are happy to provide some insights into how results from GRAPHIC could be used for, e.g., curriculum learning. While GRAPHIC does not provide details on individual samples, the approach can be used to get some **notion of class difficulty**.

---

> > ### Author Response · Authors · 2025-12-17
> > **Response to Reviewer uBCv (part 2)**
> >
> > We have added a new analysis in **Appendix A.12** that explicitly studies class difficulty over training. There, we analyze the evolution of class out-degree in the confusion graphs for the classes that are most and least confused by the converged model. We also find that some of these apparently difficult classes are already identified as ambiguous.
> >
> > Prior work [1] constructs auxiliary tasks by organizing classes into hierarchical stages, moving from coarse to fine labels, and uses CMs to identify this hierarchy. Similarly, GRAPHIC identifies CCs that reflect how classes are grouped by the model at different stages of training. These communities could directly be interpreted as intermediate learning targets of varying difficulty.
> >
> > In this sense, GRAPHIC could be used, following this approach, to define tasks based on the CCs of an NN. So one could first train on CCs observed at early epochs, then progressively refine these communities using later graphs, and finally transition to individual classes.
> >
> > **We have added Appendix A.12 to show how the out-degree evolves over the training.**
> >
> > >(stronger) I am unsure how insightful the linear separability study is. Previous work has discussed that certain classes are more separable in certain layers, but I do not see how actionable that insight is.
> >
> > We understand the reviewer's concern about actionability of the insights on linear separability. While we do not act on our results, they may be of interest to the community. Auxiliary classifiers are for example utilized in order to improve NN performance [2][3]. However, our results suggest that these methods may not transfer to transformers: in our experiments, linear separability in early transformer layers decreases after an initial increase. This also suggests that methods enforcing linear separability in early layers may actually decrease performance of transformers. This may caution practitioners.
> >
> > > (stronger) I did not fully understand the purpose of the distillation loss. It will cause some of the geometry learned in higher layers to be embedded into the lower layers, so the resulting linear classifiers may not be a good representation of the learned embedding up to that layer.
> >
> > We thank the reviewer for raising this point and giving us the opportunity to clarify. In our manuscript we discuss the **influence of the parameter $\lambda$** in **Appendix A.3**. Our experiments, however, focus on training with either the ground truth labels or the predicted labels of the NN. Training on the ground truth is straight forward. For the predicted labels, we set $\lambda=0$, meaning the LCs are trained only on the model's predictions to learn about the model's internal state. While one could argue that this introduces knowledge from later layers into earlier layers, we view this as a feature: it allows us to probe how much information from higher layers is already present in the representations of early layers and what information can already be derived from these early layers. This gives insight into the progression of learning and the internal state of the network throughout training.
> >
> > >(stronger) The loss in (4) does not seem to include a regularization term, which might cause some issues related to the uniqueness of the solution of the linear classifier.
> >
> > We appreciate the thorough feedback. While training the LCs in our main experiments does not include a regularization term, we added experiments **in Appendix A.4** on using **weight decay** and found that it does not meaningfully affect our results. We are, however, happy to **add regularization as an option in our code** for other users.
> >
> >
> > >groups have index c, vertices have index i. This could be confusing, given that classes are usually index with c
> >
> > We apologize for the confusion and have **adjusted this in the revised manuscript**.
> >
> > > In (2): maybe write $\sum_{i\in\mathbb{G}u} \sum{j\in\mathbb{G}v} C^\text{ad}{i,j}$
> >
> > We thank the reviewer for this suggestion and have **revised the manuscript** accordingly.
> >
> > > in (1), the adjacency matrix is using the wrong symbol, it should be bold
> >         if you meant to use the non-bold version to identify as a scalar, it conflicts with the number of classes C
> >
> > We understand that $C^\text{ad}_{i,j}$ may seem counterintuitive as matrices are generally denoted in bold, however, we are following the notation provided by TMLR (https://github.com/goodfeli/dlbook_notation/), where this is suggested. To ensure this is not confused with the number of classes, we have **changed the variable to $M$ in the revised manuscript**.
> >
> > >(Alain et al., 2017) seems to omit Yoshua Bengio in the reference
> >
> > We apologize sincerely for this oversight and thank the reviewer for pointing this out. **The reference has been corrected in the revised manuscript.**
> >
> > >distillation (Hinton et al., 2015) could be cited for the distillation loss
> >
> > We thank the reviewer for this comment and **have updated the paper**.

---

> > > ### Author Response · Authors · 2025-12-17
> > > **Response to Reviewer uBCv (References)**
> > >
> > > [1] Otilia Stretcu, Emmanouil Antonios Platanios, Tom Mitchell, and Barnabás Póczos. Coarse-to-fine curriculum learning for classification. In International Conference on Learning Representations (ICLR) Workshop on Bridging AI and Cognitive Science (BAICS), 2020.
> > >
> > > [2] Yakoub Bazi, Mohamad M Al Rahhal, Haikel Alhichri, and Naif Alajlan. Simple yet effective fine-tuning of deep CNNs using an auxiliary classification loss for remote sensing scene classification. Remote Sensing, 11(24):2908, 2019.
> > >
> > > [3] Liu, Yishu and Liu, Yingbin and Ding, Liwang. Scene classification based on two-stage deep feature fusion. IEEE Geoscience and Remote Sensing Letters, 15(2):183–186, 2018.

---

### Author Response · Authors · 2025-12-17
**Global Response**

We thank all reviewers for their thoughtful and constructive feedback. We are encouraged that all reviewers agree that the claims in the paper are supported by clear and convincing evidence, and that the findings are of interest to the TMLR audience. In particular, the reviewers highlighted the following strengths:
* **Novel methodology and perspective on training dynamics**, often overlooked by post-hoc approaches. (Dp2K, uBCv)
* **Clear and well-structured writing**, with effective sectioning, informative appendices, and helpful “Main Takeaway” summaries. (Dp2K, uBCv)
* **Depth of experimental analysis** supported by specific examples. (Dp2K)
* **Valuable insights from human study** to verify claims made in the paper. (uBCv)
* **Transformers and CNNs** under evaluation as approach is architecture-agnostic. (pw9o, uBCv)

Alongside their positive feedback, the reviewers provided insightful questions and constructive criticisms. We have addressed each comment individually, and for clarity, **all changes (except appendix labeling) in the revised manuscript are highlighted in blue.**

Here, we summarize the major changes in the paper:
* A **comparison to previous methods** has been added in **Appendix A.1**. (pw9o, uBCv)
* Training with **weight decay** is now addressed in **Appendix A.4**. (uBCv)
* **Scalability** is now addressed in **Appendix A.5**. (Dp2K)
* **Robustness of LC training** is now dicussed in more detail in **Appendix A.3 and A.6**. (pw9o)
* **Computational Overhead and Guidelines for Practitioners** are dicussed in **Appendices A.7 and A.8** of the revised script, respectively. (Dp2K, pw9o)
* A discussion on the **evolution of graph sparsity** has been added in **Appendix A.10**. (pw9o)
* **Class difficulty** is now discussed in **Appendix A.12**. (uBCv)

Once again, we thank the reviewers as well as the action editor for the time and effort they have put and are putting into our work.

Sincerely,
GRAPHIC authors

---

### Author Response · Authors · 2026-01-19
**Camera-Ready Version**

Dear Action Editor and Reviewers,

We have now uploaded the camera-ready version of the manuscript. We would like to once again sincerely thank the reviewers and the action editor for their valuable feedback and input.

Sincerely,
GRAPHIC Authors

---

### Author Response · Authors · 2026-05-11
**Minor Correction**

While extending this work, we noticed that our implementation of the weighted assortativity coefficient differed from the definition given in Newman (2003), which we cite. The numerical impact on the reported values is small. All findings, conclusions, and interpretations of the paper remain unchanged.
The updated PDF revises:
  - Equation (2) on p. 5;
  - Figure 3 (p. 9) and Figures 44 and 45 (p. 39), recomputed with the corrected formula;
  - The accompanying paragraph in Appendix A.13, which now reflects the updated random-grouping baseline.

We thank the editors in chief for their guidance. Keeping the released code, the paper text, and the cited reference in agreement is important for reproducibility, and we are grateful to be able to post this correction.

---

### Decision · Action_Editor_hZp3 · 2026-01-13

**Recommendation:** Accept as is

**Audience:**

Yes

**Audience Explanation:**

The approach goes beyond existing methods, notably by considering the evolution of class separability over training time and resolved across layers. The authors provide evidence that this additional information leads to novel insights compared to post-hoc analysis of the confusion matrix. All referees agreed on the interest of the paper for the community working on explainability.

**Claims And Evidence:**

Yes

**Claims Explanation:**

The authors focus on image data to explore and validate the proposed GRAPHIC approach. These experiments comprise medium-sized CNNs and vision transformers (ResNet-50 and EffVit) and both CIFAR-100 and TinyImagenet datasets. Also a human annotation experiment is performed to analyze the type of confusions made by humans compared to the models.

Referees Dp2K and uBCv where very positive about the clarity, depth and quality of the analysis. All referees felt that their main concerns were addressed in the revision.